# MI-TRQR: Mutual Information-Based Temporal Redundancy Quantification and Reduction for Energy-Efficient Spiking Neural Networks

**Dengfeng Xue[1][*] Wenjuan Li[2,3][†] Yifan Lu[2,3] Chunfeng Yuan[2,3,4] Yufan Liu[2,3] Wei Liu[2,3]**
**Man Yao[2] Li Yang[2,3] Guoqi Li[2] Bing Li[2,3,4,5] Stephen Maybank[6] Weiming Hu[2,3,4,7] Zhetao Li[8]**
[1]School of Computer Science and Technology, Xidian University
[2]State Key Laboratory of Multimodal Artificial Intelligence Systems, CASIA
[3]Beijing Key Laboratory of Super Intelligent Security of Multi-Modal Information
[4]School of Artificial Intelligence, University of Chinese Academy of Sciences [5]PeopleAI Inc.
[6]Department of Computer Science and Information Systems, Birkbeck College, University of London
[7]School of Information Science and Technology, ShanghaiTech University
[8]College of Information Science and Technology, Jinan University

## Abstract

Brain-inspired spiking neural networks (SNNs) provide energy-efficient computation through event-driven processing. However, the shared weights across multiple timesteps lead to serious temporal feature redundancy, limiting both efficiency and performance. This issue is further aggravated when processing static images due to the duplicated input. To mitigate this problem, we propose a parameter-free and plug-and-play module named Mutual Information-based Temporal Redundancy Quantification and Reduction (MI-TRQR), constructing energy-efficient SNNs. Specifically, Mutual Information (MI) is properly introduced to quantify redundancy between discrete spike features at different timesteps on two spatial scales: pixel (local) and the entire spatial features (global). Based on the multi-scale redundancy quantification, we apply a probabilistic masking strategy to remove redundant spikes. The final representation is subsequently recalibrated to account for the spike removal. Extensive experimental results demonstrate that our MI-TRQR achieves sparser spiking firing, higher energy efficiency, and better performance concurrently with different SNN architectures in tasks of neuromorphic data classification, static data classification, and time-series forecasting. Notably, MI-TRQR increases accuracy by **1.7%** on CIFAR10-DVS with 4 timesteps while reducing energy cost by **37.5%**. Our codes are available at `https://github.com/dfxue/MI-TRQR`.

## 1 Introduction

Spiking Neural Networks (SNNs), inspired by the processing mechanisms of biological neural networks [18], offer an energy-efficient approach to neuromorphic hardware by performing spike-based accumulation and avoiding the computation of zero-value inputs (i.e., event-driven) [36, 48, 10]. With the release of neuromorphic chips [5, 37] and the proposal of algorithms for various tasks [56, 11, 87, 79, 50, 63, 73, 64, 55], neuromorphic computing systems are progressing toward practical deployment in real-world applications [30].

SNNs rely on sequential timesteps to transmit temporal information, and their outputs are typically averaged across time to improve accuracy [8, 77]. The temporally shared weights induce spatio-

---

[*]An intern at CASIA.
[†]Corresponding author: wenjuan.li@ia.ac.cn

39th Conference on Neural Information Processing Systems (NeurIPS 2025).

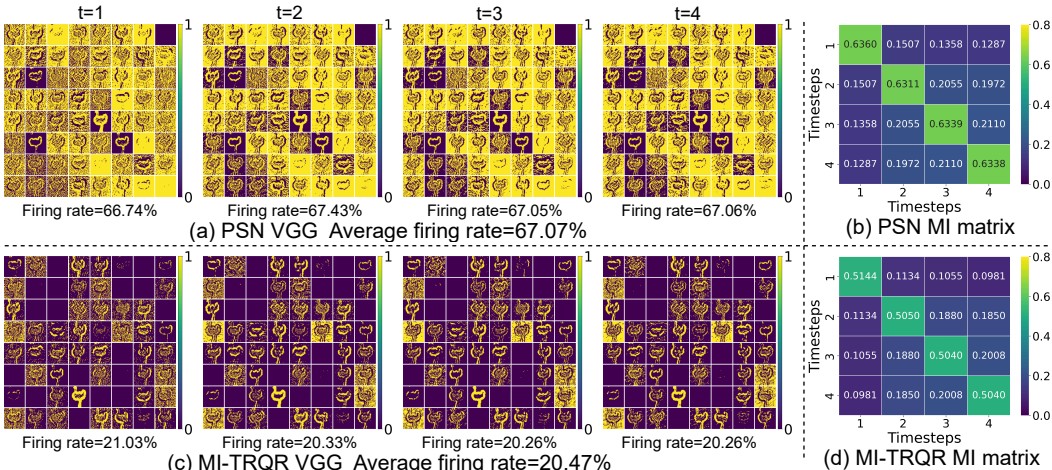

Figure 1: Visualization of spike features on CIFAR10-DVS: (a) Spike features in PSN [14] and (b) their MI matrix; (c) Spike features in MI-TRQR and (d) their MI matrix. MI-TRQR reduces the spiking firing rate by 69.48% (67.07% → 20.47%) and temporal redundancy by about 24% (computed from the redundancy reduction between the first timestep and other timesteps, e.g., 0.1507→0.1134).

temporal invariance [22], introducing serious feature redundancy and resulting in a high spiking firing rate [68, 25], as shown in Fig. 1(a). This phenomenon degrades energy efficiency and compromises the compactness of learned features. When processing static images, existing methods often duplicate the same input across all timesteps [12, 20], further exacerbating temporal redundancy in SNNs [68]. This practice has been shown to impair both the energy efficiency and overall performance of SNNs [48, 38, 69].

Energy efficiency has been a longstanding concern in recent research [54, 49, 46, 4], motivating the research and development of diverse strategies aimed at reducing redundancy. Kim et al. [25] found that features at later timesteps had minimal impact on the final predictions, highlighting considerable temporal redundancy. Yao et al. [68] conducted a systematic yet qualitative analysis on redundancy. Qin et al. [44] provided a quantitative yet indirect strategy by defining similar spikes through the cosine similarity of their membrane potential. We argue that it is insufficient to quantify the redundancy between the spike features due to two intrinsic faults: (1) the inability of a linear tool to capture correlations between floating-point membrane potentials, and (2) the quantization error caused by the discontinuous and nonlinear property of the spiking activation function. The absence of direct redundancy quantification significantly hinders progress toward highly energy-efficient SNNs.

To directly and accurately quantify redundancy between discrete spike features, we propose to use Mutual Information (MI), which is a principled and widely used metric [43, 81, 82, 57]. Existing metrics, such as Pearson correlation and Euclidean distance, measure basic similarity but fall short in capturing the complex and non-linear similarity between high-dimensional spike features [40]. In contrast, Mutual Information (MI) captures complex statistical dependencies through probability distribution analysis, making it well-suited for quantifying redundancy between the high-dimensional discrete spike features. In this work, we calculate MI between spike features at different timesteps, as shown in Fig. 1(b) and (d).

Previous studies analyzing neural recordings from various monkey brain regions have shown that spike features are significantly less redundant with activity-dependent depression [15]. Predictive coding has also demonstrated the ability to learn efficient visual representations by removing pixel-wise redundant spikes [2, 39]. Inspired by these works, we propose a parameter-free, plug-and-play MI-based Temporal Redundancy Quantification and Reduction (MI-TRQR) module, which is seamlessly integrated into the SNNs. Specifically, MI is used to quantify temporal redundancy between high-level spike features on pixel-level (local) and the entire spatial scale (global). These multi-scale redundancy quantifications are aggregated to compute a pixel-wise probability for removing spikes, which is achieved using a binary mask. Existing SNNs typically produce final representations by averaging outputs across timesteps, whether for classification [9, 12, 20] or forecasting [35],

implicitly assuming a uniform temporal contribution or information distribution. However, due to the non-uniform information distribution after spike removal, we recalibrate the weights for the final representation. As illustrated in Fig. 1, MI-TRQR reduces spiking firing rate and temporal redundancy. The proposed MI-TRQR is validated on neuromorphic data classification, static image classification, and time-series forecasting tasks. Experimental results demonstrate that our MI-TRQR obtains more compact representations, improving both performance and energy efficiency. Our contributions are as follows:

- We use MI to directly and accurately quantify temporal redundancy between high-dimensional discrete spike features at multiple spatial scales in SNNs.

- We propose a parameter-free, plug-and-play MI-TRQR module, which removes pixel-wise redundant spikes based on the multi-scale redundancy quantification. The weight of the final representation is recalibrated to balance the information distribution.

- We demonstrate the significant advantages of our approach by comparing the energy consumption of MI-TRQR with baseline methods, showing a clear improvement. Extensive experiments across a range of tasks confirm that our method achieves higher accuracy and enhanced energy efficiency.

## 2   Related Work

Redundancy is a critical factor that must be addressed when designing efficient SNNs. In the following, we briefly review some representative studies that focus on analyzing and/or reducing redundancy in SNNs.

### 2.1   Redundancy Analysis

Kim et al. [25] investigated the temporal information distribution and identified the Temporal Information Concentration (TIC) phenomenon, wherein information is highly concentrated in the early timesteps after training. This observation highlights the presence of serious temporal redundancy. Yao et al. [68] conducted a systematic analysis, particularly for event-based vision tasks. They discussed three key questions ('which', 'why', and 'how') and concluded that redundancy primarily stems from the spatio-temporal invariance caused by temporally shared weights in SNNs [22]. Qin et al. [44] introduced the concept of a spike cluster, defined based on the cosine similarity of membrane potentials across timesteps. Despite substantial progress in redundancy-related research in recent years, direct quantification of redundancy between spike features across timesteps remains largely unexplored. In this paper, we address this gap by using MI to directly quantify the redundancy between spike features at different timesteps.

### 2.2   Redundancy Reduction

Many methods have been proposed to mitigate redundancy in different ways. Perez et al. [42] introduced an early sparse backpropagation algorithm tailored for SNNs. Subsequent works further applied sparsity regularization during backpropagation to enhance the energy efficiency of SNNs [66, 75]. A recent series of studies [72, 71, 67, 68, 65] presented a cohesive exploration of attention mechanisms in SNNs. These studies used diverse attention designs to strategically modulate membrane potential distributions and spiking responses across various dimensions, such as spatial, temporal, channel-wise, and/or their combinations. They underscored the transformative impact of attention mechanisms in SNNs, leading to more efficient and accurate models. However, such attention modules inevitably increase network complexity and introduce additional multiplications, which may raise hardware costs and hinder deployment on resource-limited edge devices. The temporal self-erasing method dynamically adjusted the regions of interest for different timesteps [34]. Another widely adopted approach is to apply penalty functions to encourage the deep network to learn sparser spike representations [41, 74, 27, 78]. These methods often require careful tuning of many hyperparameters, which may increase the training complexity. In this paper, we develop a parameter-free and plug-and-play module to enable SNNs to learn compact and powerful representations.

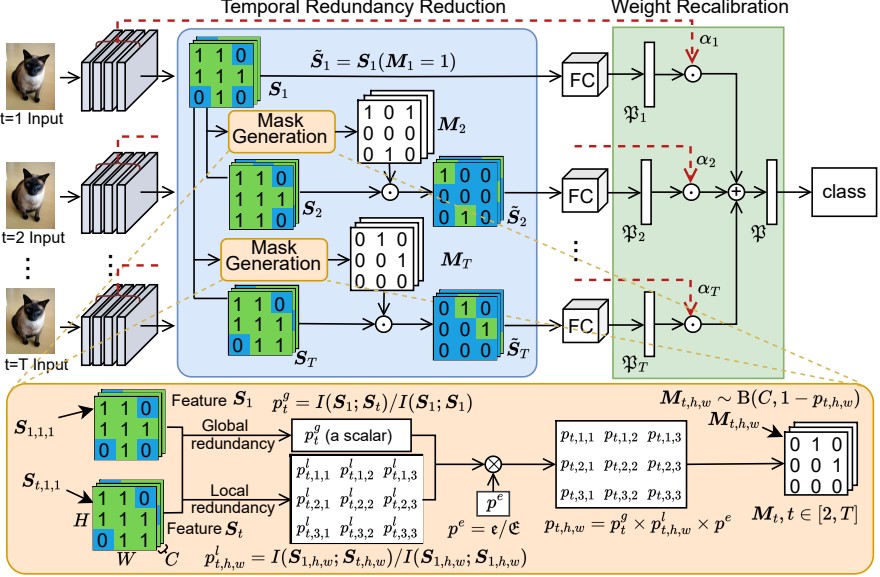

Figure 2: Our MI-TRQR module preserves the original spike features at the first timestep. At subsequent timesteps, local and global redundancy quantification are first aggregated with a training factor. This aggregation is then used to compute a probability that guides the mask generation. The generated mask is responsible for removing pixel-wise redundant spikes, aiming to achieve temporal redundancy reduction. Weight recalibration balances the non-uniform information distribution.

# 3 Methodology

In this section, we introduce the proposed MI-TRQR module, which is integrated after the final stage to process the high-level features [83], as shown in Fig. 2. Section 3.1 details how MI is used to quantify temporal redundancy between spike features. Section 3.2 introduces the temporal redundancy reduction strategy, removing redundant spikes using the multi-scale redundancy quantification. Section 3.3 recalibrates the weight of the final representation based on the information density.

## 3.1 Temporal Redundancy Quantification

MI is particularly effective for evaluating similarity between discrete variables, as elaborated in Appendix A. Therefore, MI is a suitable metric to quantify temporal redundancy between binary spike features. Given the four-dimensional binary spike feature $S \in \{0,1\}^{T \times C \times H \times W}$ where $T, C, H, W$ indicate the timestep, channel, height, and width respectively, we define two terms to quantify temporal redundancy.

**Definition 1.** *Global redundancy* quantifies the overall temporal redundancy between spike features at two different timesteps and is denoted as $R^g$. It is computed as follows:

$$R^g(i,j) = I(S_i; S_j) = \sum_{s_i \in S_i} \sum_{s_j \in S_j} p(s_i, s_j) \log \left( \frac{p(s_i, s_j)}{p(s_i)p(s_j)} \right), \tag{1}$$

where $I(S_i; S_j)$ denote MI between spiking features $S_i, S_j \in \{0,1\}^{C \times H \times W}$ at timestep $i$ and $j$. $s_i$ and $s_j$ represent specific values of $S_i$ and $S_j$. $p(s_i)$ and $p(s_j)$ are the marginal probability distributions of $s_i$ and $s_j$, respectively. $p(s_i, s_j)$ is their joint probability distribution. $R^g(i,j) \geq 0$ is a float scalar that quantifies the degree of *global redundancy* between spike features $S_i$ and $S_j$.

**Definition 2.** *Local redundancy* quantifies temporal redundancy at the pixel scale. For each spatial location, we compute the MI between the local spike vectors at two timesteps:

$$R^l_{h,w}(i,j) = I(\boldsymbol{S}_{i,h,w}; \boldsymbol{S}_{j,h,w}) = \sum_{s_i \in \boldsymbol{S}_{i,h,w}} \sum_{s_j \in \boldsymbol{S}_{j,h,w}} p(s_i, s_j) \log \left( \frac{p(s_i, s_j)}{p(s_i)p(s_j)} \right), \quad (2)$$

where $\forall h \in [1, H], \forall w \in [1, W]$. $\boldsymbol{S}_{i,h,w}, \boldsymbol{S}_{j,h,w} \in \{0,1\}^C$ are the raw pixel (spatial location $(h, w)$) at timestep $i$ and $j$, respectively. $R^l_{h,w}(i,j) \geq 0$ is also a float scalar, indicating the pixel-wise redundancy between $\boldsymbol{S}_{i,h,w}$ and $\boldsymbol{S}_{j,h,w}$. It corresponds to the $(h, w)$-th element of the *local redundancy* matrix $R^l(i,j) \in \mathbb{R}^{H \times W}$.

## 3.2 Temporal Redundancy Reduction

Based on the redundancy quantification between spike features at multiple spatial scales, we derive a probability that estimates the likelihood of each pixel-wise spike being redundant. This probability is then used to generate a pixel-wise temporal redundancy mask, which selectively removes redundant spikes.

**Temporal redundancy-guided probability derivation.** The *global redundancy* between spike features $\boldsymbol{S}_1$ and $\boldsymbol{S}_t$ ($t \in [2, T]$) is served as the global probability factor for spikes removal, denoted by $p^g_t = R^g(1, t)$. We observe that the MI of a feature with itself $I(\boldsymbol{S}_1, \boldsymbol{S}_1)$ is not equal to one. This is evident from the varying diagonal elements in the MI matrix shown in Fig. 1. Therefore, we normalize the global probability factor $p^g_t$ as follows:

$$p^g_t = \frac{R^g(1, t)}{R^g(1, 1)} = \frac{I(\boldsymbol{S}_1; \boldsymbol{S}_t)}{I(\boldsymbol{S}_1; \boldsymbol{S}_1)}, \quad (3)$$

where $p^g_t \in [0, 1)$, since $I(\boldsymbol{S}_1; \boldsymbol{S}_t) < I(\boldsymbol{S}_1; \boldsymbol{S}_1)$ when $t \neq 1$, as shown in the Fig 1(b).

Similarly to the computation of $p^g_t$ in Eq. 3, the local probability factor $p^l_{t,h,w}$ is defined as follows:

$$p^l_{t,h,w} = \frac{R^l_{h,w}(1, t)}{R^l_{h,w}(1, 1)} = \frac{I(\boldsymbol{S}_{1,h,w}; \boldsymbol{S}_{t,h,w})}{I(\boldsymbol{S}_{1,h,w}; \boldsymbol{S}_{1,h,w})}, \quad (4)$$

where $p^l_{t,h,w} \in [0, 1)$, similar to the $p^g_t$.

In addition, the TIC phenomenon demonstrates that information gradually concentrates on the first timestep as training goes on [25]. Accordingly, we retain the spikes at the first timestep, as illustrated in Fig. 2. However, when removing spikes, it is crucial to consider the training epoch, as the informative spikes are still distributed across the later timesteps during the early training stages. Thus, we introduce a training-dependent factor $p^e$ into the pixel-wise probability $p_{t,h,w}$:

$$p_{t,h,w} = p^g_t \times p^l_{t,h,w} \times p^e, \quad p^e = \mathfrak{e}/\mathfrak{E}, \quad (5)$$

where $\mathfrak{e}$ and $\mathfrak{E}$ denote the current training epoch and the total number of epochs, respectively.

**Temporal redundancy-based mask.** We use the pixel-wise probability $p_{t,h,w}$ to generate a binary mask at timestep $t$:

$$\boldsymbol{M}_t = \begin{cases} \{1\}^{C \times H \times W} & \text{if } t = 1 \\ [\boldsymbol{M}_{t,h,w}], & \text{otherwise} \end{cases}, \quad \boldsymbol{M}_{t,h,w} \sim \mathrm{B}(C, 1 - p_{t,h,w}), \quad (6)$$

where $\boldsymbol{M}_t \in \{0, 1\}^{C \times H \times W}$ denotes the mask at timestep $t$. $\boldsymbol{M}_{t,h,w} \in \{0, 1\}^C$ represents the pixel at spatial location $(h, w)$. $\mathrm{B}(\cdot, \cdot)$ denotes the Binomial distribution. The spike removal operation using the mask $\boldsymbol{M}_t$ is expressed as:

$$\tilde{\boldsymbol{S}}_t = \boldsymbol{S}_t \odot \boldsymbol{M}_t, \quad (7)$$

where $\tilde{\boldsymbol{S}}_t$ denotes the output spike features at timestep $t$, and $\odot$ is the element-wise multiplication. The mask at the first timestep, $\boldsymbol{M}_1$, consists entirely of one, thereby preserving the original spikes.

---

**Algorithm 1** The implementation of removing pixel-wise redundant spikes

---

1: **Input:** Spike feature $S \in \{0,1\}^{T \times C \times H \times W}$ , the training factor $p^e$
2: **Output:** Spike feature $\tilde{S} \in \{0,1\}^{T \times C \times H \times W}$
3: Obtain the number of spatial pixels: $N = H \times W$
4: Flatten spatial dimension: $S' = \text{reshape}(S, (T, C, N))$
5: Define a matrix filled with 1 for redundancy quantification: $\Re' \leftarrow \text{ones}(T, N)$
6: **for** $t = 1 \rightarrow T$ **do**
7:     Compute *global redundancy*: $R^g(1,t) = I(S'_1; S'_t)$
8:     **for** $n = 1 \rightarrow N$ **do**
9:         Compute *local redundancy*: $R^l_n(1,t) = I(S'_{1,n}; S'_{t,n})$,
10:         Get the pixel-wise combined redundancy metric: $\Re'[t,n] = R^g(1,t) \times R^l_n(1,t)$
11:     **end for**
12: **end for**
13: Unfold spatial dimension: $\Re = \text{reshape}(\Re', (T, H, W))$
14: Obtain a $T \times N$ probability matrix: $P = \Re[:,:,:]/\Re[0,;,:] \times p^e$
15: Set the probability at the first timestep into zeros: $P[0]=0$
16: Generate mask: $M = [M_{t,h,w}], \quad M_{t,h,w} \sim \text{B}(C, 1 - P[t,h,w])$,
17: Get the output spike feature with element-wise multiplication: $\tilde{S} = S \odot M$

---

For $t \in [2, T]$, zero values in $M_t$ convert active spikes (1) into inactive states (0). Consequently, a proportion $(p_{t,h,w})$ of spikes at the pixel $S_{t,h,w}$ is removed. The pseudo-code for spike removal is given in Alg. 1. Spike removal also reduces the spiking firing rate in earlier layers, which is analyzed in Appendix B.

### 3.3 Weight Recalibration

The spike removal at later timesteps introduces a significant divergence in information distribution across timesteps. Consequently, it is unsuitable to simply average temporal outputs for the final representation, as done in existing works [12, 20, 35]. To address this issue, we adaptively recalibrate the weight of the final representation using the normalized network spiking firing rate:

$$\alpha_t = \frac{fr_t^{net}}{\sum_{\tau=1}^{T} fr_\tau^{net}}, \quad fr_t^{net} = \frac{\sum_{l=1}^{L} N_{l,t}^s}{\sum_{l=1}^{L} N_{l,t}^e}, \tag{8}$$

where $a_t$ and $fr_t^{net}$ denote the weight and network spiking firing rate at timestep $t$, respectively. $N_{l,t}^s$ indicates the number of spikes, and $N_{l,t}^e$ indicates the total number of elements, both at timestep $t$ and layer $l$. $L$ is the total number of layers. In this way, the final representation $\mathfrak{P}$ is obtained as follows:

$$\mathfrak{P} = \sum_{t=1}^{T} (\alpha_t \times \mathfrak{P}_t), \tag{9}$$

where $\mathfrak{P}_t$ denotes the representation at timestep $t$.

## 4 Experiments

In this section, we report the experimental results for MI-TRQR on both classification and time-series forecasting tasks. The operation of removing spikes is only used during training. The experimental setup is described in Appendix C.1.

### 4.1 Comparison with Other Methods

**Neuromorphic data classification.** On CIFAR10-DVS, MI-TRQR consistently surpasses PSN [14] in both accuracy and energy efficiency across different timesteps, as shown in Tab. 1. When $T$=4, PSN vs. **MI-TRQR**: Accuracy, 82.3% vs. **83.9%(+1.6%)**; Power, 0.72mJ vs. **0.45mJ (-0.27mJ,**

Table 1: Comparison with other methods on neuromorphic datasets. Power denotes the energy cost of inference one test sample. ↑ and ↓ indicate desired metric direction. Results denoted by ♠ were obtained through our reimplementation.

| Dataset | Model | Network | T | Accuracy (↑%) | Power (↓mJ) |
|---------|-------|---------|---|---------------|-------------|
| CIFAR10-DVS | DeepTAGE [33] | VGG-11 | 10 | 81.2 | - |
| | TIM [53] | Spikformer | 10 | 81.6 | - |
| | SEMM [85] | Spikformer-2-256 | 16 | 82.9 | - |
| | RM-SNN [71] | VGG | 10 | 82.9 | - |
| | STAtten [29] | SpikingReformer-4-384 | 16 | 83.9 | - |
| | QKFormer [84] | HST-2-256 | 16 | 84.0 | - |
| | CLIF [21] | VGG | 10 | 86.1 | - |
| | PSN [14] | VGG | 4 | 82.3 | 0.72 |
| | | | 8 | 85.3 | 1.28 |
| | | | 10 | 85.9 | 1.20 |
| | MI-TRQR (ours) | VGG | 4 | **83.9**±0.05 | **0.45**±0.06 |
| | | | 8 | **86.2**±0.12 | **0.74**±0.07 |
| | | | 10 | **86.5**±0.13 | **0.84**±0.07 |
| Gait | ASA [68] | 3-Layer SNN | 10 | 83.2 | - |
| | 3D GCN [60] | - | 1 | 86.0 | - |
| | DSNN [70] | 4-Layer SNN | - | 90.2 | - |
| | PSN [14] | 3B-Net | 10 | 88.8♠ | 0.19 |
| | MI-TRQR (ours) | 3B-Net | 10 | **90.6**±0.13 | **0.13**±0.08 |

Table 2: Comparison with other methods on ImageNet.

| Method | Model | Network | T | Accuracy (↑%) | Power (↓mJ) |
|--------|-------|---------|---|---------------|-------------|
| ANN2SNN | Hybird [47] | ResNet34 | 250 | 61.48 | - |
| | Tandem [62] | VGG-16 | 16 | 65.08 | - |
| | Two-stage [61] | ResNet34 | 16 | 67.77 | - |
| | Optimal [3] | ResNet34 | 32 | 69.37 | - |
| Direct Training | OSR+OTS [89] | ResNet34 | 4 | 67.54 | - |
| | EnOF-SNN [17] | ResNet34 | 4 | 67.40 | - |
| | $rate_M$ [76] | MS-ResNet34 | 4 | 70.01 | - |
| | GAC-SNN [45] | MS-ResNet34 | 6 | 70.42 | 3.38 |
| | IMP+LTS [52] | SEW-ResNet50 | 4 | 71.83 | 3.11 |
| | PSN [14] | SEW-ResNet18 | 4 | 67.63 | 2.42 |
| | | SEW-ResNet34 | 4 | 70.54 | 3.70 |
| | | SEW-ResNet50 | 4 | 72.01♠ | 4.11 |
| Direct Training | MI-TRQR (ours) | SEW-ResNet18 | 4 | **68.28**±0.12 | **1.92**±0.07 |
| | | SEW-ResNet34 | 4 | **71.06**±0.11 | **3.06**±0.09 |
| | | SEW-ResNet50 | 4 | **73.23**±013 | **3.09**±0.08 |

↓**37.50%**). Similar advantages of our MI-TRQR are observed when $T$=8,10. MI-TRQR shows 1.8% accuracy improvement and 31.6% energy reduction over PSN on Gait ($T$=10).

**Static data classification.** On ImageNet, MI-TRQR consistently outperforms PSN across different backbones, as shown in Tab. 2. For example, with ResNet50, PSN vs. **MI-TRQR**: Accuracy, 72.01% vs. **73.23% (+1.22%)**; Power, 4.11mJ vs. **3.09mJ (-1.02mJ, ↓ 24.82%)**. More experiments on CIFAR10/100 are provided in Appendix C.2.1. On CIFAR10, PSN vs. **MI-TRQR**: Accuracy, 95.32% vs. **95.83% (+0.51%)**; Power, 1.32mJ vs. **0.35mJ (-0.97mJ, ↓ 73.48%)**.

**Time-series forecasting.** In Tab. 3, we show the superiority of our MI-TRQR over CPG-PE [35] in two metrics ($R^2$ and RSE) on the Electricity dataset with various prediction lengths 6,24,48,96.

Table 3: Results on Electricity. Our results are averaged across 3 random seeds.

| Method | Metric | Eletricity | | | | Avg. |
|---|---|---|---|---|---|---|
| | | 6 | 24 | 48 | 96 | |
| Spikformer [86] | $R^2\uparrow$ | .956 | .955 | .953 | .943 | .952 |
| | $RSE\downarrow$ | .371 | .375 | .386 | .450 | .396 |
| CPG-PE [35] | $R^2\uparrow$ | .971 | .971 | .968 | .962 | .968 |
| | $RSE\downarrow$ | .304 | .308 | .311 | .439 | .341 |
| MI-TRQR (ours) | $R^2\uparrow$ | **.973** | **.972** | **.972** | **.967** | **.971** |
| | $RSE\downarrow$ | **.292** | **.307** | **.297** | **.368** | **.316** |

Table 4: Impact of *local redundancy* (LR) and *global redundancy* (GR) on CIFAR10-DVS.

| Method | LR | GR | T | Accuracy (↑%) | Firing rate (↓%) | Training time |
|---|---|---|---|---|---|---|
| PSN [14] | - | - | 4 | 82.3 | 14.47 | **24.1**s |
| MI-TRQR (ours) | ✓ | - | 4 | 83.2 (↑0.9) | 9.73 (↓32.76) | 51.1s |
| | - | ✓ | 4 | 83.4 (↑1.1) | 10.91 (↓24.60) | 24.8s |
| | ✓ | ✓ | 4 | **84.0** (↑1.7) | **8.35** (↓42.29) | 51.9s |

Table 5: Impact of MI-TRQR placement on CIFAR10-DVS.

| Method | Placement | T | Accuracy(↑%) | Firing rate(↓%) | Training time |
|---|---|---|---|---|---|
| PSN [14] | - | 4 | 82.3 | 14.47 | **24.1**s |
| MI-TRQR(ours) | After layer 4 | 4 | 82.8(↑0.5) | 11.52(↓20.39%) | 597.3s |
| | After layer 6 | 4 | 83.4(↑1.1) | 10.18(↓29.65%) | 171.6s |
| | After layer 6+8 | 4 | 83.5(↑1.2) | 9.74(↓32.69%) | 222.6s |
| | After layer 8(last) | 4 | **84.0**(↑1.7) | **8.35**(↓42.29%) | **51.9**s |

## 4.2 Ablation Study

We conducted ablation studies on CIFAR10-DVS to verify the impact of weight recalibration and the effectiveness, efficiency, and optimal placement of the MI-TRQR module. Additional ablation results on CIFAR10 are provided in Appendix C.2.2. More ablation studies are provided in Appendix C.3.

**Effectiveness.** In Tab. 4, we quantitatively show the gains of incorporating local and/or global redundancy for learning more compact representations. By leveraging multi-scale redundancy, MI-TRQR obtains the best accuracy of 84.0%(+1.7%). We also report the training time per epoch to provide a comprehensive comparison.

**Efficiency.** In Tab. 4, we can see that using multi-scale redundancy reduces the spiking firing rate by 42.29% (from 14.47% to 8.35%). The detailed spike counts and firing rates are presented in Fig. 3. Key observations are: (1) MI-TRQR consistently fires fewer spikes than PSN at each timestep; (2) its spike removal propagates from deeper layers to shallower layers; (3) it consistently reduces the spiking firing rate across various datasets (CIFAR10 in Appendix C.2.2).

**MI-TRQR placement.** In Tab. 5, we report the results when MI-TRQR is integrated after different convolutional layers. Results show that positioning MI-TRQR after the last stage (layer 8) yields the biggest advantages in both accuracy and firing rate compared to PSN, confirming the benefit of targeting MI-TRQR to high-level features. In addition, the computation of probability $p_{t,h,w}$ in Eq. 5 is spatiotemporally dependent, with a computational complexity of $O(T)$ for global redundancy and $O(T \times H \times W)$ for local redundancy. Inserting the module into earlier layers leads to a higher computational cost. We provide the corresponding time of training an epoch.

**Weight recalibration.** Tab. 6 presents the impact of weight recalibration on CIFAR10-DVS. MI-TRQR with reweighting yields the highest accuracy while consuming the least energy.

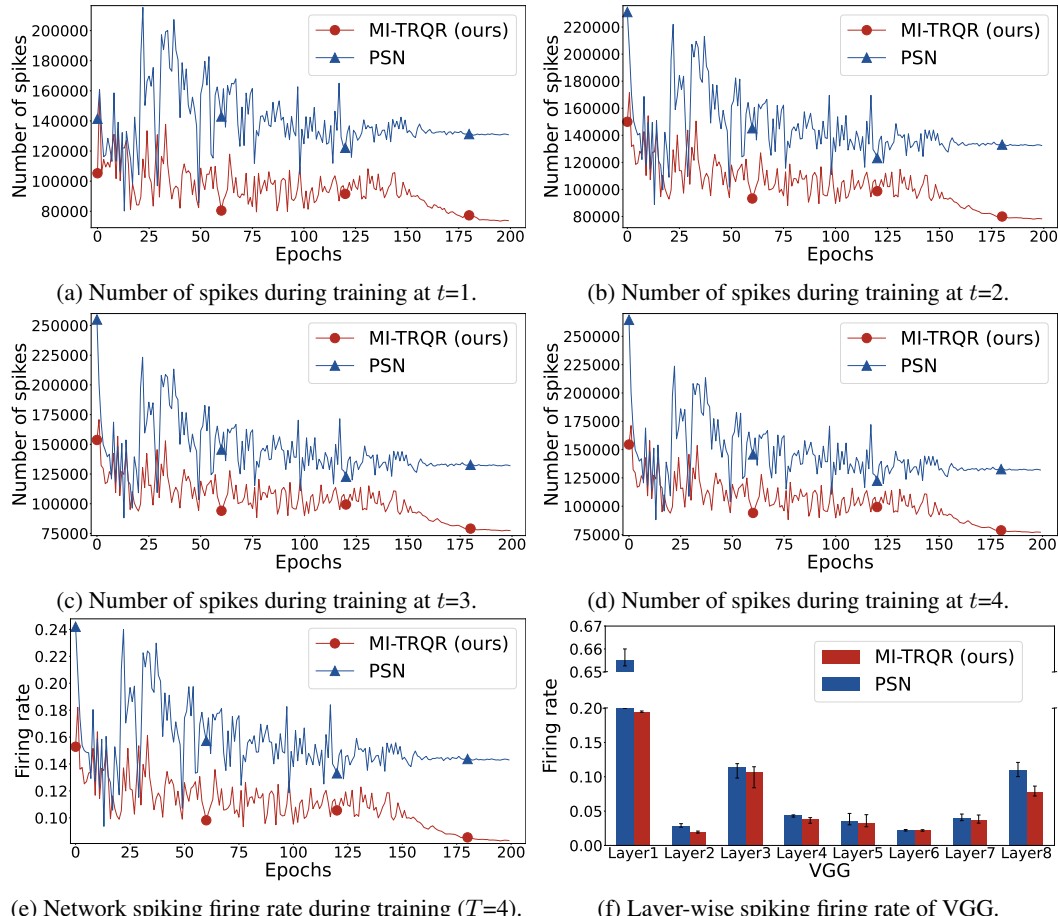

(a) Number of spikes during training at $t$=1.

(b) Number of spikes during training at $t$=2.

(c) Number of spikes during training at $t$=3.

(d) Number of spikes during training at $t$=4.

(e) Network spiking firing rate during training ($T$=4).

(f) Layer-wise spiking firing rate of VGG.

Figure 3: Comparison of the number of spikes and firing rate in different methods on CIFAR10-DVS.

Table 6: Impact of weight recalibration on CIFAR10-DVS.

| Method | Reweighting | T | Accuracy ($\uparrow$%) | Firing rate ($\downarrow$%) |
|---|---|---|---|---|
| PSN [14] | - | 4 | 82.3 | 14.47 |
| MI-TRQR (ours) | - | 4 | 83.4($\uparrow$1.1) | 9.22($\downarrow$36.28) |
| | $\checkmark$ | 4 | **84.0**($\uparrow$1.7) | **8.35**($\downarrow$42.29) |

## 4.3 Visualization

**CIFAR10-DVS.** In Fig. 1, we can observe that MI-TRQR reduces the spiking firing rate by 69.48% (67.07% $\to$ 20.47%) and temporal redundancy by approximately 24% (e.g., 0.1507 $\to$0.1134) even at a shallow layer.

**ImageNet.** In Fig. 4, we visualize spike features and their MI matrices. We can see that MI-TRQR significantly decreases the temporal redundancy between spike features by 79% (e.g.,) and reduces the average spiking firing rate by 18.39% (23.34% vs. 28.60%). These results demonstrate that MI-TRQR can effectively remove redundant spikes and enhance representation compactness.

## 5 Conclusion and Limitations

In this paper, we propose an effective and efficient module, named MI-TRQR. Based on the temporal redundancy quantification using MI, MI-TRQR identifies and removes pixel-wise redundant spikes, enabling SNNs to learn more compact and powerful representations. Experimental results on various

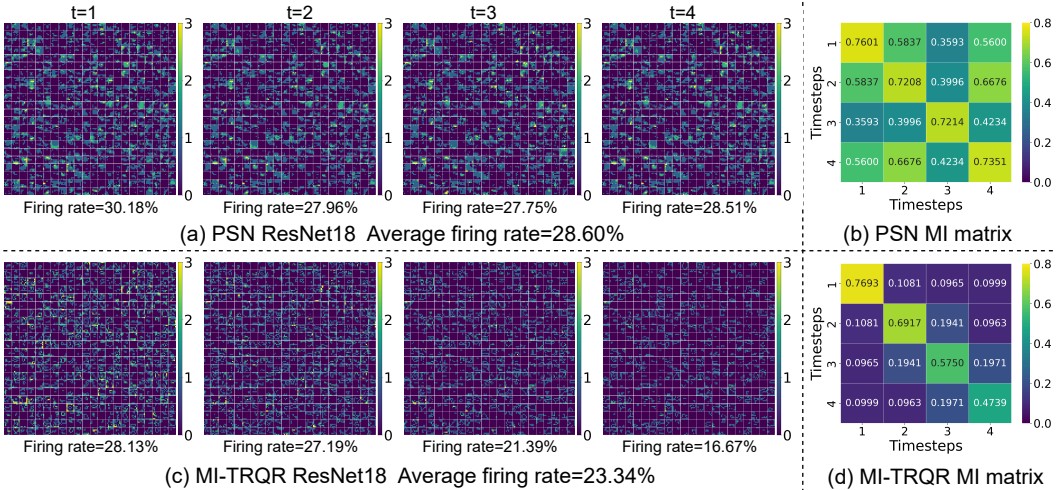

Figure 4: Visualization of spike features after stage 4 of ResNet18 on ImageNet: (a) Spike features in PSN and (b) their MI matrix; (c) Spike features in MI-TRQR and (d) their MI matrix. For a single sample, the spike features has shape $[T, C, H, W]$ (timestep, channel, height, width). Since $T = 4$, we first split it into four $[C, H, W]$ features (shown on the left). We plot each channel in a small grid. We report the average spiking firing rates under the spike features. MI-TRQR reduces the spiking firing rate by 18.39% (28.60% $\rightarrow$ 23.34%) and temporal redundancy by about 79% (computed from the redundancy reduction between the first timestep and other timesteps, e.g., 0.5837$\rightarrow$0.1081)

tasks demonstrate that MI-TRQR consistently outperforms baseline methods in both accuracy and energy efficiency. To the best of our knowledge, this is the first study to use MI to quantify redundancy between spike features in SNNs, providing a foundation for subsequent related research.

**Limitations** Although our method does not introduce extra cost during inference, it requires computing mutual information between spike features during training, resulting in increased computational cost and longer training time.

## Acknowledgements

This work is supported by the Natural Science Foundation of China (62036011, 62192782, W2411053, 62032020, U2441241, U24A20331, 62372451, 62192785, 62372082, 62202469, 62403462, 62202470, 62473363), Beijing Natural Science Foundation (L223003, L243015, L251005, JQ24022), the Key R&D Program of Xinjiang Uyghur Autonomous Region (2023B01005), CAAI-Ant Group Research Fund (CAAI-MYJJ 2024-02), and Young Elite Scientists Sponsorship Program by CAST (2024QNRC001).

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

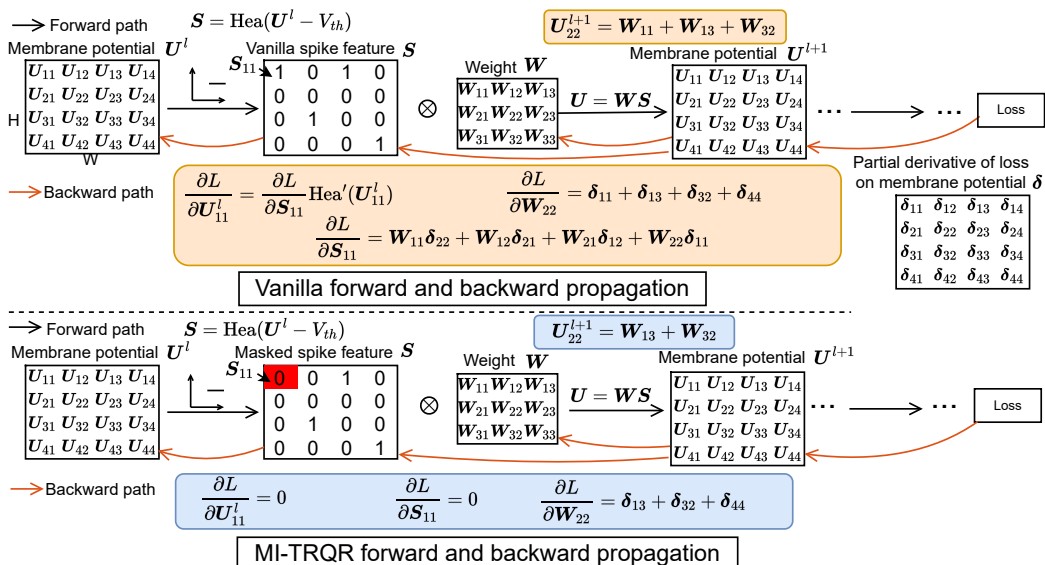

Figure 5: Forward and backward process in vanilla and MI-TRQR method. We detail the forward and backward propagation processes of a layer of the SNN network, only considering the calculation in the spatial domain. When performing convolution calculations, the padding (without drawing) and stride are set to 1. A masked spike $S_{11}$ is marked in red. The differences in the computing formulations between different methods are enclosed by boxes of various colors.

## A    Mutual Information

Mutual information (MI), first formalized in Shannon's information theory [51], provides a generalized dependence measure between random variables. Unlike measures limited to linear relationships, MI captures complex statistical dependencies through probability distribution analysis. This property makes it suitable for quantifying similarity between discrete spike features in SNNs. The MI between two spike features $\boldsymbol{S}$ and $\boldsymbol{S}'$ is expressed as:

$$I(\boldsymbol{S}; \boldsymbol{S}') = \sum_{s \in \boldsymbol{S}} \sum_{s' \in \boldsymbol{S}'} p(s, s') \log \left( \frac{p(s, s')}{p(s)p(s')} \right) \tag{10}$$

where $s$ and $s'$ indicate the values in feature $S$ and $S'$, respectively. $p(s, s')$ is the joint probability distribution, and $p(s)$ and $p(s')$ are the marginal probability distributions. MI satisfies two fundamental properties: (1) Non-negativity ($I(\boldsymbol{S}; \boldsymbol{S}') \geq 0$) ensures meaningful information measures, and a higher MI value indicates greater similarity. $I(\boldsymbol{S}; \boldsymbol{S}') = 0$, if and only if $\boldsymbol{S}$ and $\boldsymbol{S}'$ are independent random variables. (2) Symmetry ($I(\boldsymbol{S}; \boldsymbol{S}') = I(\boldsymbol{S}'; \boldsymbol{S})$) reflects bidirectional dependence.

## B    Analysis of MI-TRQR

### B.1    Analysis of Forward Propagation

SNNs become a promising alternative to ANNs, contributing to their advantages in energy efficiency. Unlike ANNs, SNNs achieve inter-layer communication with binary signals (0-nothing or 1-spike) [36]. As a kind of neuromorphic computing algorithm, SNNs only compute sparse spike-based accumulation (AC) and avoid handling the zero-value inputs [48], such as the calculation of membrane potential $\boldsymbol{U} = \boldsymbol{W}\boldsymbol{S}$, as shown in Fig. 5. The energy consumption of SNNs $E_{Conv}$ is influenced by the spiking firing rates of input features. Referring to the equation in [45], the inference energy consumption of a network is computed as follows:

$$E_{net} = E_{MAC} \cdot FL_{Conv}^1 + E_{AC} \cdot T \cdot \left( \sum_{n=2}^{N} FL_{Conv}^n \cdot fr_{Conv}^n + \sum_{m=1}^{M} FL_{FC}^m \cdot fr_{FC}^m \right) \quad (11)$$

where $FL_{Conv}^n$, $FL_{FC}^m$, $fr_{Conv}^n$, and $fr_{FC}^m$ are the FLOPs and spiking firing rate of the $n$-th convolution and $m$-th FC layer. $N$ and $M$ are the total number of convolution and FC layers. $E_{MAC}$ and $E_{AC}$ are the energy consumption of MAC and AC operation. Following the works [74, 19], we assume that $E_{MAC} = 4.6pJ$ and $E_{AC} = 0.9pJ$.

In forward propagation, our MI-TRQR removes some spikes. We can see the effect on the calculation of membrane potential $\boldsymbol{U}_{22}^{l+1}$:

$$\text{Vanilla: } \boldsymbol{U}_{22}^{l+1} = \boldsymbol{W}_{11} + \boldsymbol{W}_{13} + \boldsymbol{W}_{32} \quad (12)$$

$$\text{MI-TRQR: } \boldsymbol{U}_{22}^{l+1} = \boldsymbol{W}_{13} + \boldsymbol{W}_{32} \quad (13)$$

where $l$ indicates the layer number. We can see that MI-TRQR requires less accumulation than the vanilla method.

## B.2   Analysis of Backward Propagation

In the backward propagation, we discuss the partial derivative of loss on different variables with different methods in the following three cases. Firstly, we denote the partial derivative of loss on the $l + 1$-th layer membrane potential $\boldsymbol{U}^{l+1}$ as:

$$\boldsymbol{\delta} = \frac{\partial L}{\partial \boldsymbol{U}^{l+1}} \quad (14)$$

where $\boldsymbol{\delta}$ is the partial derivative, as shown in Fig. 5.

**Case 1: The partial derivative of loss on weight.** Similar to ANNs, the partial derivative is:

$$\frac{\partial L}{\partial \boldsymbol{W}_{m,n}} = \sum_{i=1}^{H} \sum_{j=1}^{W} \boldsymbol{\delta}_{i,j} \boldsymbol{S}_{i+m-2,j+n-2} \quad (15)$$

where $H$ and $W$ indicate the height and width of features. The difference is that the spike feature $\boldsymbol{S}$ is a sparse binary tensor, meaning the computing of Eq. 15 is spike-based accumulation. For example, with different methods, the partial derivative of loss on $\boldsymbol{W}_{22}$ is:

$$\text{Vanilla: } \frac{\partial L}{\boldsymbol{W}_{22}} = \boldsymbol{\delta}_{11} + \boldsymbol{\delta}_{13} + \boldsymbol{\delta}_{32} + \boldsymbol{\delta}_{44} \quad (16)$$

$$\text{MI-TRQR: } \frac{\partial L}{\boldsymbol{W}_{22}} = \boldsymbol{\delta}_{13} + \boldsymbol{\delta}_{32} + \boldsymbol{\delta}_{44} \quad (17)$$

We can observe that the gradient calculation of MI-TRQR requires less accumulation than the vanilla method, meaning the parameter update in MI-TRQR removes the redundant/invalid gradient in the vanilla method.

**Case 2: The partial derivative of loss on spike features** is computed with:

$$\frac{\partial L}{\partial \boldsymbol{S}_{m,n}} = \sum_{i=1}^{k} \sum_{j=1}^{k} \boldsymbol{W}_{i,j} \boldsymbol{\delta}_{m-i+2,n-j+2} \quad (18)$$

where $k$ indicates the size of the convolution kernel. However, the calculation of this partial derivative has a precondition: The input feature needs to involve the forward calculation, which means that the input feature must be non-zero. In other words, the gradient of the zero input is zero. Thus, the partial derivative of loss on the spike $\boldsymbol{S}_{11}$ is:

$$\text{Vanilla: } \frac{\partial L}{\boldsymbol{S}_{11}} = \boldsymbol{W}_{11}\boldsymbol{\delta}_{22} + \boldsymbol{W}_{12}\boldsymbol{\delta}_{21} + \boldsymbol{W}_{21}\boldsymbol{\delta}_{12} + \boldsymbol{W}_{22}\boldsymbol{\delta}_{11} \tag{19}$$

$$\text{MI-TRQR: } \frac{\partial L}{\boldsymbol{S}_{11}} = 0 \tag{20}$$

We can see that the partial derivative of loss on the removed spike in the two methods is completely different. Thus, we perceive that removing a spike means removing its derivative/gradient.

**Case 3: The partial derivative of loss on the $l$-th layer membrane potential $\boldsymbol{U}^l$** is easily obtained based on case 2. This partial derivative can be represented as:

$$\frac{\partial \boldsymbol{L}}{\partial \boldsymbol{U}^l} = \frac{\partial \boldsymbol{L}}{\partial \boldsymbol{S}}\frac{\partial \boldsymbol{S}}{\partial \boldsymbol{U}^l} = \frac{\partial \boldsymbol{L}}{\partial \boldsymbol{S}}\text{Hea}'(\boldsymbol{U}^l) \tag{21}$$

where $\text{Hea}'(\boldsymbol{U}^l)$ indicates the derivative of $\text{Hea}(\boldsymbol{U}^l)$. Specifically, with different methods, the partial derivative of loss on the membrane potential $\boldsymbol{U}^l_{11}$ is:

$$\text{Vanilla: } \frac{\partial L}{\boldsymbol{U}^l_{11}} = \frac{\partial \boldsymbol{L}}{\partial \boldsymbol{S}_{11}}\text{Hea}'(\boldsymbol{U}^l_{11}) \tag{22}$$

$$\text{MI-TRQR: } \frac{\partial L}{\boldsymbol{U}^l_{11}} = 0 \times \text{Hea}'(\boldsymbol{U}^l_{11}) = 0 \tag{23}$$

We can see that the removed spike can influence the gradient of the membrane potential at the previous layer.

**Summary.** Through the above formula derivation and analysis, we can conclude that removing the spikes after the last convolutional layer can affect the derivative of the previous membrane potential, and further affect the weight gradient and parameter updates at the previous layers. In this way, MI-TRQR can learn compact and powerful representations in SNNs.

## C  Experiments

### C.1  Experimental Setup

**Datasets.**

CIFAR10/100 [26], two small datasets, contain 50,000 training and 10,000 test samples. ImageNet [6] is a large-scale dataset of about 1.3 million images (1.25 million for training and 0.5 million for testing) across 1,000 classes. CIFAR10-DVS [31], an event-stream dataset converted from CIFAR10 with dynamic vision sensors, includes 10,000 event streams in 10 classes. DVS128 Gait [59] contains 4200 samples in 20 classes from 21 volunteers. Electricity [28] captures hourly electricity consumption measured in kilowatt-hours (kWh).

**Implementation.** We adopt the PSN [14] and STMixer [7] as baselines of the classification task. We adopt the CPG-PE [35] as the baseline of the time-series forecasting task. We follow their experimental setup, such as network architecture, training methods, data preprocessing, etc. All experiments were conducted on a Ubuntu 20.04.6 LTS server with 8 NVIDIA GeForce RTX 3090. The MI dependence is calculated with the TorchMetrics package.

For DVS128 Gait, we designed a small 3B-Net. Its structure is c128k3s1-BN-PLIF-SEW Block-MPk2s2*3-FC20. Here, c128k3s1 denotes a convolution layer with channels 128, kernel size 3, and stride 1. PLIF refers to the PLIF spiking neuron [13]. MPk2s2 represents the max pooling with kernel size 2 and stride 2. SEW Block is a custom-designed SEW block, expressed as c32k3s1-BN-PLIF-c128k3s1-BN-PLIF. The symbol *3 indicates that the structure in  is repeated three times.

Table 7: Comparison with other methods on CIFAR10/100.

| Dataset | Method | Backbone | T | Accuracy (↑%) | Power (↓mJ) |
|---|---|---|---|---|---|
| CIFAR10 | TAB [24] | ResNet19 | 4 | 94.76 | - |
| | RevSFormer [80] | RevSFormer-4-384 | 4 | 95.34 | - |
| | TCJA-SNN [88] | ResNet18 | 4 | 95.60 | - |
| | PSN [14] | Modified PLIF Net | 4 | 95.32 | 1.32 |
| | MI-TRQR (Ours) | Modified PLIF Net | 4 | **95.83**±0.10 | **0.35**±0.05 |
| | STATA [90] | Spikingformer | 4 | 95.8 | - |
| | STAtten [29] | SDT-2-512 | 4 | 96.03 | - |
| | STMixer [7] | STMixer-4-384-32 | 4 | 96.01 | 0.95 |
| | MI-TRQR (Ours) | STMixer-4-384-32 | 4 | **96.64**±0.12 | **0.51**±0.08 |
| CIFAR100 | SLT-TET [1] | ResNet19 | 4 | 75.01 | - |
| | NDOT$_A$ [23] | VGG11 | 4 | 76.18 | - |
| | LietE-SNN [32] | - | 6 | 77.10 | - |
| | PSN♠ [14] | ResNet18 | 4 | 75.75 | 0.43 |
| | | ResNet19 | 4 | 76.14 | 1.78 |
| | MI-TRQR (Ours) | ResNet18 | 4 | **76.70**±0.12 | **0.31**±0.08 |
| | | ResNet19 | 4 | **77.70**±0.10 | **1.24**±0.07 |
| | ST [16] | ST-4-384 | 4 | 79.69 | - |
| | SNN-ViT [58] | SDT | 4 | 80.1 | - |
| | SDT+SEMM [85] | SDT | 4 | 80.26 | - |
| | STMixer [7] | STMixer-4-384-32 | 4 | 81.87 | 1.08 |
| | MI-TRQR (Ours) | STMixer-4-384-32 | 4 | **83.06**±0.11 | **0.77**±0.08 |

Table 8: Impact of *local redundancy* (LR) and *global redundancy* (GR) on CIFAR10.

| Method | LR | GR | T | Accuracy (↑%) | Firing rate (↓%) |
|---|---|---|---|---|---|
| PSN [14] | - | - | 4 | 95.32 | 16.39 |
| MI-TRQR (ours) | ✓ | - | 4 | 95.53 (↑0.21) | 10.59 (↓35.39) |
| | - | ✓ | 4 | 95.77 (↑0.45) | 8.98 (↓45.21) |
| | ✓ | ✓ | 4 | **95.83** (↑0.51) | **5.36** (↓67.30) |

## C.2 Experimental Results on CIFAR10/100

### C.2.1 Comparison with other methods

**CIFAR10.** Tab. 7 presents the classification performance and energy consumption of various methods on CIFAR-10. MI-TRQR achieves **95.83%** accuracy, outperforming PSN (95.32%) by +0.51% while reducing energy consumption by **73.48%** (**0.35mJ** vs. 1.32mJ). With the STMixer-4-384-32 backbone, MI-TRQR attains **96.64%** accuracy, surpassing STMixer (96.01%) by 0.63% while reducing energy consumption by **46.3%** (**0.51 mJ** vs. 0.95 mJ).

**CIFAR100.** As shown in Tab. 7, MI-TRQR consistently outperforms PSN across both ResNet18 and ResNet19 backbones, achieving **76.70%** and **77.70%** accuracy respectively, compared to 75.75% and 76.14% of PSN, while also reducing power consumption from 0.43 mJ and 1.78 mJ to **0.31 mJ** and **1.24 mJ**. With the STMixer-4-384-32 backbone, MI-TRQR attains **83.06%** accuracy, outperforming the original STMixer (81.87%) by **1.19%** while reducing energy consumption by **28.70%** (**0.77 mJ** vs. 1.08 mJ).

### C.2.2 Ablation Study on CIFAl10

We conduct ablation studies on CIFAR10 to validate the effectiveness of our redundancy designs in MI-TRQR. As summarized in Tab. 8, both local and global redundancy contribute to performance

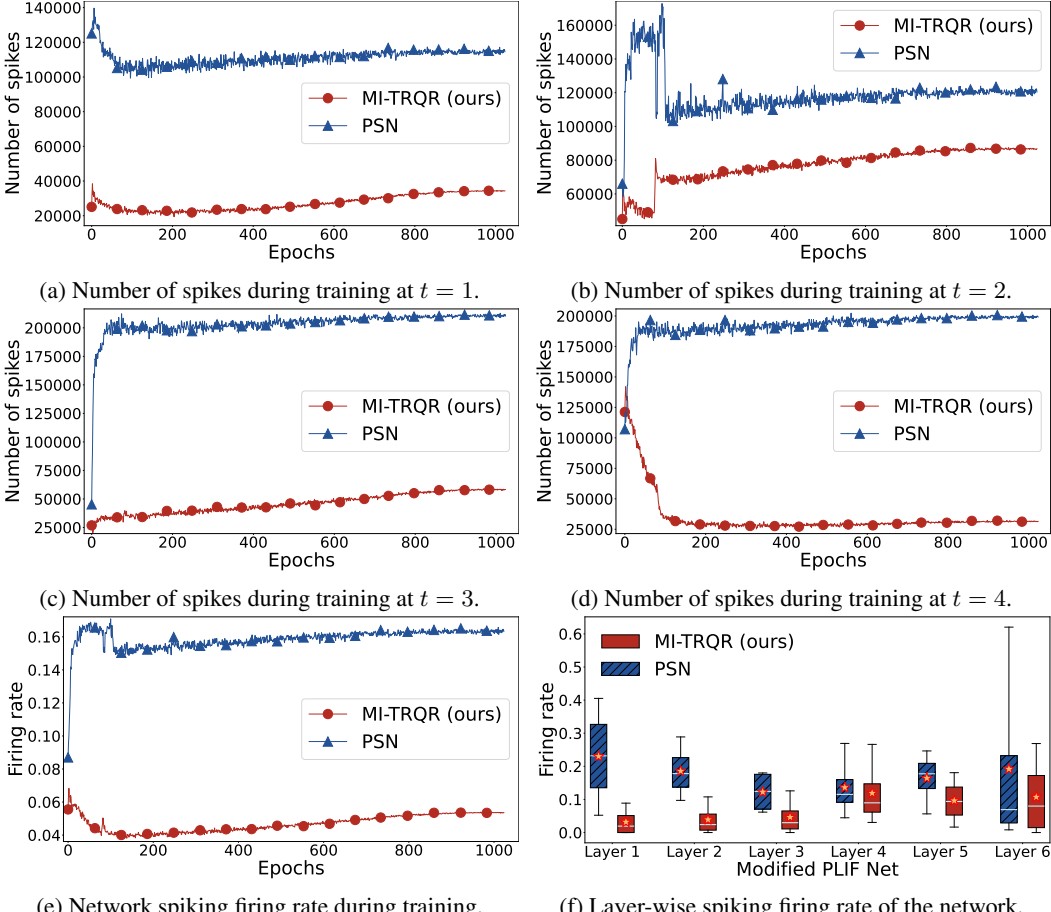

(a) Number of spikes during training at $t = 1$.

(b) Number of spikes during training at $t = 2$.

(c) Number of spikes during training at $t = 3$.

(d) Number of spikes during training at $t = 4$.

(e) Network spiking firing rate during training.

(f) Layer-wise spiking firing rate of the network.

Figure 6: Comparison of the number of spikes and firing rate in different methods on CIFAR10.

improvements over the PSN [14]. Using local redundancy alone improves accuracy by 0.21% while reducing the firing rate by 35.39%. Feature redundancy demonstrates stronger effects, achieving a 0.45% accuracy gain with 45.21% fewer spikes. Notably, combining both mechanisms yields the best results - our full model attains 95.83% accuracy (+0.51%) with only 5.36% firing rate, representing a 67.30% reduction in spike activity. The number of spikes and firing rates in various methods are shown in Fig. 6. Specifically, the number of spikes at four timesteps during training is shown from Fig. 6a to 6d. It can be seen that our MI-TRQR consistently fired fewer spikes than PSN at each timestep. The overall spiking firing rates during training are plotted in Fig. 6e. The firing rate of our MI-TRQR is reduced by 67.29% compared to that of PSN (5.36% vs. 16.39%). The layer-wise firing rates of the Modified PLIF Net are presented in Fig. 6f. Our MI-TRQR has a lower firing rate than PSN at every layer of the Modified PLIF Net.

## C.3 Extended Ablation Studies

### C.3.1 Spike normalization

To verify the effect of spike normalization in Eq. 5, we apply identical removal ratios to all spikes. Specifically, after obtaining the mask $M_t$ with $p_{t,,h,w}$, we shuffle its elements to generate a random mask for spike removal. Results in Tab. 9 indicate that both the MI-based ratio and the removed spikes play essential roles.

In Tab. 10, we evaluate different static penalty factors that increase with timesteps and serve as a regularizer on the number of spikes. The results show that these static penalties effectively suppress redundant spikes and improve overall performance. Notably, MI-TRQR achieves higher accuracy with a lower firing rate.

Table 9: Effect of a random mask on CIFAR10-DVS when $T$=4.

| Method | Shuffle | $T$ | Accuracy($\uparrow$%) | Firing rate($\downarrow$%) |
|---|---|---|---|---|
| PSN [14] | - | 4 | 82.3 | 14.47 |
| MI-TRQR (ours) | $\checkmark$ | 4 | 83.4 ($\uparrow$1.1) | 8.51 ($\downarrow$41.19) |
| | - | 4 | **84.0** ($\uparrow$1.7) | **8.35** ($\downarrow$42.29) |

Table 10: Effect of static penalty factors on CIFAR10-DVS when $T$=4.

| Method | $t=1$ | $t=2$ | $t=3$ | $t=4$ | Accuracy($\uparrow$%) | Firing rate($\downarrow$%) |
|---|---|---|---|---|---|---|
| PSN [14] | - | - | - | - | 82.3 | 14.47 |
| Static penalty | 0 | 0.1 | 0.2 | 0.3 | 83.0 ($\uparrow$0.7) | 10.83 ($\downarrow$25.16) |
| | 0 | 0.15 | 0.3 | 0.45 | 82.9 ($\uparrow$0.6) | 9.55 ($\downarrow$34.00) |
| | 0 | 0.2 | 0.4 | 0.6 | 82.8 ($\uparrow$0.5) | 10.61 ($\downarrow$26.68) |
| MI-TRQR (ours) | - | - | - | - | **84.0** ($\uparrow$1.7) | **8.35** ($\downarrow$42.29) |

Table 11: Effect of different techniques on CIFAR10-DVS when $T$=4.

| Method | Technique | $T$ | Accuracy($\uparrow$%) | Firing rate($\downarrow$%) |
|---|---|---|---|---|
| PSN [14] | - | 4 | 82.3 | 14.47 |
| Absolute distance | Euclidean | 4 | 82.4 ($\uparrow$0.1) | 11.74 ($\downarrow$18.87) |
| Linear similarity | Pearson | 4 | 82.5 ($\uparrow$0.2) | 10.92 ($\downarrow$24.53) |
| Feature direction | Cosine | 4 | 82.9 ($\uparrow$0.6) | 9.59 ($\downarrow$33.72) |
| MI-TRQR (ours) | MI | 4 | **84.0** ($\uparrow$1.7) | **8.35** ($\downarrow$42.29) |

Table 12: Effect of different factors on CIFAR10-DVS when $T$=4.

| Method | Factor $a$ | $T$ | Accuracy($\uparrow$%) | Firing rate($\downarrow$%) |
|---|---|---|---|---|
| PSN [14] | - | 4 | 82.3 | 14.47 |
| MI-TRQR (ours) | 0.5 | 4 | 84.5 ($\uparrow$2.2) | 10.66 ($\downarrow$26.33) |
| | 1 | 4 | 84.0 ($\uparrow$1.7) | 8.35 ($\downarrow$42.29) |
| | 2 | 4 | 82.7 ($\uparrow$0.4) | 8.16 ($\downarrow$43.61) |

### C.3.2 Other techniques

We replace MI with simpler similarity measures, including Euclidean similarity (via 1/(1+Euclidean distance)), Pearson correlation, and Cosine similarity. As shown in Tab. 11, spiking firing rates still decrease with these alternatives, while MI-TRQR achieves higher accuracy and a lower firing rate.

### C.3.3 Scaling factor

We multiply $p_{t,h,w}$ (Eq. 5) with an additional factor $a$, and evaluate the results under different setting in Tab. 12. As shown, the factor $a$ influences the trade-off between accuracy and firing rate.

### C.3.4 Mask generation strategies

In our method, we adopt Bernoulli-based masks to remove redundant spikes in a data-dependent manner, without introducing additional hyperparameters. This design also admits a clear theoretical interpretation. To further validate its effectiveness, we compare it with a global redundancy-based thresholding approach for pixel-wise spike removal. Specifically, for the feature $S_t \in \{0,1\}^{C \times H \times W} (t >$ 1), we compute the global redundancy $p_t^g$ (Eq. 3) and the local redundancy $p_{t,h,w}^l$ (Eq. 4). If $p_{t,h,w}^l > p_t^g$, the spike feature $S_{t,h,w} \in \{0,1\}^C$ is set into 0. As shown in Tab. 13, MI-TRQR with Bernoulli-based masks achieves better performance.

Table 13: Effect of mask generation strategies on CIFAR10-DVS when $T$=4.

| Method | Mask type | $T$ | Accuracy($\uparrow$%) | Firing rate($\downarrow$%) |
|---|---|---|---|---|
| PSN [14] | - | 4 | 82.3 | 14.47 |
| MI-TRQR (ours) | Global threshold | 4 | 82.6 ($\uparrow$0.3) | 8.97 ($\downarrow$38.01) |
| | Bernoulli | 4 | **84.0** ($\uparrow$1.7) | **8.35** ($\downarrow$42.29) |

