# OpenReview forum: "MI-TRQR: Mutual Information-Based Temporal Redundancy Quantification and Reduction for Energy-Efficient Spiking Neural Networks"
_NeurIPS.cc/2025/Conference — NeurIPS 2025 poster_

### Official Review · Reviewer_wE4j · 2025-06-27

**Clarity:** 2
**Significance:** 2
**Originality:** 3
**Rating:** 3
**Confidence:** 5

**Summary:**

This paper proposes a Mutual Information-based Temporal Redundancy Quantification and Reduction (MI-TRQR) module that reduces the computational overhead imposed by spikes. This method computes global and local mutual information in spiking neural networks (SNNs) to mask redundant spikes. The authors demonstrate through experimentation that the proposed method effectively reduces the number of spikes and improves performance on object classification and time-series forecasting tasks.

**Questions:**

See weaknesses.

**Ethical Concerns:**

["NO or VERY MINOR ethics concerns only"]

**Final Justification:**

Although the author responded to my concerns, these responses did not convince me. I do not believe that this paper has reached the standard for acceptance, and therefore I am maintaining my score.

**Limitations:**

The authors claim that their method has no limitations.

**Paper Formatting Concerns:**

No formatting issues.

**Quality:**

2

**Strengths And Weaknesses:**

Strength:
1. The proposed MI-TRQR module does not require additional attention mechanisms or other learnable parameters.
2. Inserting the proposed module into the existing SNN reduces the number of spikes and improves performance.

Weakness:
1. **Theoretical analysis**. Although the authors claim that mutual information is an effective way to evaluate the similarity between discrete variables, there is a lack of theoretical analysis regarding the effectiveness of binary spikes. Does the limited range of representations of binary spikes affect the precision of mutual information?
2. **Complexity and energy efficiency**. Reducing the number of 1-value spikes in the SNN can reduce deployment power consumption by avoiding addition operations caused by those spikes. However, computing mutual information requires a large number of multiplications and logarithmic operations (Equations 1 and 2), which consume more power than addition. In fact, this increases the overall inference power and complexity rather than decreasing them. Therefore, the authors should not rely solely on the number of spikes to calculate the theoretical power consumption.
3. **Method rationality**. The authors retained all spikes from the initial time step and used them as references to calculate probabilities for subsequent time steps. However, these calculations did not consider interactions between other time steps. Furthermore, recent work [1,2] has shown that the performance of SNNs degrades dramatically when relying only on early time steps. Does this suggest that the limited amount of information in the first time step cannot be used as a reference benchmark?
4. **Method effectiveness**. The authors point out that removing spikes results in an imbalanced distribution across time steps. Therefore, they reweight the weights for each time step. However, the authors calculated the weights using firing rates only and did not adjust them for the unbalanced distribution. Even more importantly, no ablation studies have verified the effectiveness of this reweighting.
5. **Insufficient experimental comparisons**. Although the related work is described, the authors do not compare the proposed method with other lightweighting or redundancy reduction methods in the experimental section. This makes it difficult to verify whether the proposed method has any significant advantages. It is recommended that the authors add a detailed comparison with other redundancy reduction methods such as [3,4].
6. **Insignificant performance gains**. The performance improvement on the time-series forecasting task is too minimal to determine whether the method is effective. More experimental evidence is recommended. Additionally, are the results reported in the paper the best results obtained from multiple experiments, such as CIFAR10-DVS? If so, please provide the average results.

```
[1] Rethinking Spiking Neural Networks from an Ensemble Learning Perspective. ICLR. 2025.
[2] Efficient Logit-based Knowledge Distillation of Deep Spiking Neural Networks for Full-Range Timestep Deployment. ICML. 2025.
[3] Inherent Redundancy in Spiking Neural Networks. ICCV. 2023.
[4] Sparser spiking activity can be better: Feature Refine-and-Mask spiking neural network for event-based visual recognition. 2023.
```

---

> ### Author Rebuttal · Authors · 2025-07-29
>
> Thank you very much for taking the time to review our paper. Our responses to your questions are as follows:
>
> **1. Question: Theoretical analysis. Although the authors claim that mutual information is an effective way to evaluate the similarity between discrete variables, there is a lack of theoretical analysis regarding the effectiveness of binary spikes. Does the limited range of representations of binary spikes affect the precision of mutual information?**
>
> Response: Thanks for the meticulous comment.
>
> (1) Mutual information (MI) is a widely used non-linear metric for quantifying statistical dependence between discrete variables. In the context of binary spike features, where each element is assigned a value of 0 (no spike) or 1 (spike), MI remains theoretically valid and interpretable.
>
> (2) The state space of binary variables ({0,1}) does not limit the theoretical precision of MI. As established in classical information theory [1], MI measures the reduction in uncertainty of one variable given the knowledge of another. For binary variables, MI is bounded between 0 and 1, with 1 indicating identical variables and 0 indicating completely independent variables. Empirical evidence also supports the utility of MI: for example, MI increases monotonically with similarity intensity, even when applied to discrete or event-based representations [2]. In a word, the limited range of representations of binary spikes **does not affect** the precision of MI.
>
> **2. Question: Complexity and energy efficiency. Reducing the number of 1-value spikes in the SNN can reduce deployment power consumption by avoiding addition operations caused by those spikes. However, computing mutual information requires a large number of multiplications and logarithmic operations (Equations 1 and 2), which consume more power than addition. In fact, this increases the overall inference power and complexity rather than decreasing them. Therefore, the authors should not rely solely on the number of spikes to calculate the theoretical power consumption.**
>
> Response: Yes. You are right. Computing mutual information requires a large number of multiplications and logarithmic operations. Nevertheless, as stated in line 525, our module is applied solely during training and has no impact on the inference stage. Therefore, it is reasonable to rely solely on the number of spikes to calculate the theoretical energy consumption. We will include the related statement (our module is only used during training) in the main text.
>
> **3. Question: Method rationality. The authors retained all spikes from the initial time step and used them as references to calculate probabilities for subsequent time steps. However, these calculations did not consider interactions between other time steps. Furthermore, recent work [3.4] has shown that the performance of SNNs degrades dramatically when relying only on early time steps. Does this suggest that the limited amount of information in the first time step cannot be used as a reference benchmark?**
>
> Response: Thanks for your insightful and valuable thinking.
>
> (1) The interactions between other time steps.
>
> The temporal information concentration (TIC) phenomenon indicates that, as training goes on, the amount of information at the first timestep increases while that of the other timesteps decreases. Therefore, we argue that especially in the later stages of training, the first time step carries the most critical information and can be used as a reference, whereas the others become negligible and unsuitable for reference. Accordingly, we measure the mutual information between the features at each subsequent timestep and those at the first timestep. Even when mutual information is measured between features from other timesteps, the resulting values are of limited significance, since both features convey little information on their own.
>
> (2) The amount of information in the first time step.
>
> We study these two works [3.4] and find the related demonstration as follows:
>
> >We view this averaging operation as an ensemble strategy: averaging the different outputs of multiple models improves overall performance. [3]
>
> >Inspired by ensemble learning, viewing the mean output of SNNs as an ensemble aggregated through voting over time, it becomes apparent that the overall outcome across these temporal dimensions tends to improve as the accuracy at each individual timestep increases. [4]
>
> We also agree that averaging the different outputs of multiple models improves overall performance. We argue that this is not in conflict with the TIC phenomenon [5], which demonstrates that information gradually concentrates on the first timestep as training goes on. In other words, the TIC phenomenon means that the improvement extent will gradually decrease as training goes on. Moreover, to avoid misunderstanding, we will modify the description on line 26 in the final version as follows:
>
> >SNNs rely on sequential timesteps to transmit temporal information, **and their outputs are typically averaged across time to improve accuracy [3,4]**. The temporally shared weights induce spatio-temporal invariance, introducing substantial feature redundancy and resulting in a high spiking firing rate, as shown in Fig. 1(a).
>
> **4. Question: Method effectiveness. The authors point out that removing spikes results in an imbalanced distribution across time steps. Therefore, they reweight the weights for each time step. However, the authors calculated the weights using firing rates only and did not adjust them for the unbalanced distribution. Even more importantly, no ablation studies have verified the effectiveness of this reweighting.**
>
> Response: Thanks for your careful review. We conduct the ablation study to verify the effectiveness of this reweighting.
>
> | Method | Reweighting | Accuracy | Firing rate |
> | --- | --- | --- | --- |
> | PSN | - | 82.3 | 14.47 |
> | MI-TRQR | - | 83.4 | 9.22 |
> | MI-TRQR | $\checkmark$ | 84.0 | 8.35 |
>
> We can observe that the reweighting is effective. We will include the ablation study in the final version.
>
> **5. Question: Insufficient experimental comparisons. Although the related work is described, the authors do not compare the proposed method with other lightweighting or redundancy reduction methods in the experimental section. This makes it difficult to verify whether the proposed method has any significant advantages. It is recommended that the authors add a detailed comparison with other redundancy reduction methods such as ASA [6] and RM-SNN [7].**
>
> Response: I have compared our method to ASA [6] on Gait. For CIFAR10-DVS, RM-SNN obtained 82.9% with $T$=10. Our method achieved 86.5%. We will include the comparison in the final version.
>
> **6. Question: Insignificant performance gains. The performance improvement on the time-series forecasting task is too minimal to determine whether the method is effective. More experimental evidence is recommended. Additionally, are the results reported in the paper the best results obtained from multiple experiments, such as CIFAR10-DVS? If so, please provide the average results.**
>
> Response: Thanks for your careful consideration.
>
> (1) Although the performance improvement is not overwhelming, it is sufficient to demonstrate the effectiveness of our method to learn more compact and powerful representations. Because our MI-TRQR not only increases the accuracy but also **reduces the energy consumption** compared to multiple SNN baselines for various tasks. We provide much evidence in various ways.
>
> (2) We follow your advice and run three experiments. The result on CIFAR10-DVS with T=4 is 83.9$\pm$0.08.
>
> **References**
>
> [1] Thomas M. Cover, and Joy A. Thomas. Elements of information theory {(2.} ed. 2006
>
> [2] Thomas Kreuz, Florian Mormann, Ralph G. Andrzejak, Alexander Kraskov, Klaus Lehnertz, and Peter Grassberger. Measuring synchronization in coupled model systems: A comparison of different approaches. Physica D, Nonlinear Phenomena, 2007, 225(1): 29-42.
>
> [3] Yongqi Ding, Lin Zuo, Mengmeng Jing, Pei He, and Hanpu Deng. Rethinking Spiking Neural Networks from an Ensemble Learning Perspective. ICLR. 2025.
>
> [4] Chengting Yu, Xiaochen Zhao, Lei Liu, Shu Yang, Gaoang Wang, Erping Li, and Aili Wang. Efficient Logit-based Knowledge Distillation of Deep Spiking Neural Networks for Full-Range Timestep Deployment. ICML. 2025.
>
> [5] Youngeun Kim, Yuhang Li, Hyoungseob Park, Yeshwanth Venkatesha, Anna Hambitzer, and Priyadarshini Panda. Exploring temporal information dynamics in spiking neural networks. In Proceedings of the AAAI Conference on Artificial Intelligence, volume 37, pages 8308–8316, 2023.
>
> [6] Man Yao, Jiakui Hu, Guangshe Zhao, Yaoyuan Wang, Ziyang Zhang, Bo Xu, and Guoqi Li. Inherent Redundancy in Spiking Neural Networks. ICCV. 2023.
>
> [7] Man Yao, Hengyu Zhang, Guangshe Zhao, Xiyu Zhang, Dingheng Wang, Gang Cao, and Guoqi Li. Sparser spiking activity can be better: Feature Refine-and-Mask spiking neural network for event-based visual recognition. 2023.

---

> > ### Comment · Reviewer_wE4j · 2025-08-06
> > **Thanks for the author's response**
> >
> > Thanks for the author's response. Although the author responded to my concerns, these responses did not convince me. I do not believe that this paper has reached the standard for acceptance, and therefore I am maintaining my score.

---

> > > ### Author Response · Authors · 2025-08-07
> > >
> > > Thanks for your feedback. Can you tell us which part is unclear? We are glad to give a further clarification.

---

### Official Review · Reviewer_nHVz · 2025-07-03

**Clarity:** 3
**Significance:** 3
**Originality:** 3
**Rating:** 5
**Confidence:** 3

**Summary:**

This paper proposes a novel approach named “MI-TRQR” that learns sparse visual representations in a spiking neural network (SNN). This is motivated by the fact that in the context of neuromorphic processing, spikes are the main source of energy consumption. The authors characterize redundancy using the concept of mutual information and develop a framework that adaptively keeps only the less redundant spikes to learn efficient (and therefore inexpensive) visual representations. Crucially, the paper was tested on different datasets (Time-series data, CIFAR10-DVS, DVS Gait and ImageNet) with different network backbones. It achieves a competitive performance with other methods while significantly reducing energy consumption.

**Questions:**

1) I would kindly ask the authors to please clarify further how the weight recalibration happens in the network.

2) A relevant body of work which has not been discussed in the paper is the one of predictive coding [1, 2]. In predictive coding, pixel-wise redundant spikes (and internal layer spikes) are removed, usually following some sort of minimization of the so-called free-energy [1]. I believe that the authors should at the very least cite this body of work and discuss differences between their work and some of the spiking counterparts of predictive coding [2], as both types of models aim at learning efficient visual representations through the removal of redundant spikes.

3) I would kindly ask the authors to analyze the weaknesses/limitations of their work.

4) I encourage the authors to modify the formatting of Sec. 4.1, as the text is not properly justified.

[1] Bogacz, Rafal. "A tutorial on the free-energy framework for modelling perception and learning." Journal of mathematical psychology 76 (2017): 198-211.

[2] N'dri, Antony W., et al. "Predictive Coding with Spiking Neural Networks: a Survey." arXiv preprint arXiv:2409.05386 (2024).

**Ethical Concerns:**

["NO or VERY MINOR ethics concerns only"]

**Final Justification:**

The author's response addressed my concerns in terms of weaknesses and strengths of the work and also clarified the methodology of the module described in their work. I will therefore be maintaining my score.

**Limitations:**

The authors do not address the weaknesses/limitations of their work. I suggest to the authors to add a paragraph describing such limitations, and provided some examples in the "Strengths and Weaknesses" section of this review.

**Paper Formatting Concerns:**

N/A.

**Quality:**

3

**Strengths And Weaknesses:**

Strengths:

1) A new direct training model that learns efficient visual representations for spiking neural networks. The principle of the model is sound and derived from mathematical principles.
2) An evaluation on multiple datasets with convincing results that are in the same range as other methods while significantly reducing the energy consumption of the network (see Tab. 1 with MI-TRQR achieving a higher accuracy with less energy consumptions than a PSN model that uses the same network). Results are averaged over multiple simulations which highlight their statistical significance.
3) The study of the effect of local redundancy and global redundancy on accuracy (Table 4) gives an interesting perspective on how optimizing efficiency on local and global information affects information coding.

Weaknesses:

1) The addition of the MI-TRQR module, while reducing energy consumption leads to additional parameters, but the number of parameters (memory) could be a bottleneck depending on the application considered.
2) I find it unclear how the weight recalibration takes place. It appears that the input spikes are still projected through some sort of residual connections (see Fig. 2, dashed red line). Equations 8 and 9 however appear to specify otherwise, as the spiking rate (which should be, I reckon the one after spike removal) of the network is used to calculate α.
3) While the MI-TRQR module allows for the removal of input spikes, the module in itself is not spike-based as it calculates and optimizes continuous quantities, which I find unfortunate for the deployment of fully spiking-based neural network processes.

---

> ### Author Rebuttal · Authors · 2025-07-29
>
> Thank you very much for taking the time to review our paper. Our responses to your questions are as follows:
>
> **1. Question: The addition of the MI-TRQR module, while reducing energy consumption leads to additional parameters, but the number of parameters (memory) could be a bottleneck depending on the application considered.**
>
> Response: Our MI-TRQR requires extra computational and memory overhead but no additional parameters. Our MI-TRQR is only applied during training, which is stated in line 525. At the inference stage, our method **behaves identically** to the baseline model. Therefore, our method **does not pose a memory bottleneck** in various applications. We will include the related statement (our module is only used during training) in the main text.
>
> **2. Question: I find it unclear how the weight recalibration takes place. It appears that the input spikes are still projected through some sort of residual connections (see Fig. 2, dashed red line). Equations 8 and 9 however appear to specify otherwise, as the spiking rate (which should be, I reckon the one after spike removal) of the network is used to calculate α.**
>
> Response:
>
> (1) Yes. Your understanding is correct. The weight $a_t$ is computed based on the spiking firing rate of the network at the timestep $t$. Specifically, we count the number of spikes and total elements after each spiking neuron at every timestep and use this to derive the firing rate used for weight recalibration.
>
> (2) We apologize for the discontinuity of the dashed red line in Fig. 2. It should be connected to the same line in the upper left corner, indicating the network spiking firing rate. The break was introduced to avoid visual clutter in the middle part.
>
> **3. Question: While the MI-TRQR module allows for the removal of input spikes, the module in itself is not spike-based as it calculates and optimizes continuous quantities, which I find unfortunate for the deployment of fully spiking-based neural network processes.**
>
> Response: Your question is very insightful. Our MI-TRQR calculates continuous quantities. Since current neuromorphic chips still support operations on continuous values. The chips are energy-efficient as long as the outputs of spiking neurons are binary (0 or 1). Therefore, our module remains compatible with the deployment of fully spiking-based neural network processes.
>
> **4. Question: I would kindly ask the authors to please clarify further how the weight recalibration happens in the network.**
>
> Response: During training, at the $l$-th layer of spiking neuron, we count the number of spikes $N_{l,t}^{s}$ and total number of elements $N^e_{l,t}$ at each timestep. These values are used to calculate the timestep-wise firing rate $fr^{net}_t$:
>
> $$
> fr_{t}^{net} = \frac{\sum_{l=1}^{L} N_{l,t}^{s}}{\sum_{l=1}^{L} N_{l,t}^{e}}
> $$
>
> where $t$ indicates the timestep. $L$ is the total number of spiking neuron layers. $fr_{t}^{net}$ is then used to calculate the weight in Eq. 8. We will provide a detailed demonstration in the final version.
>
> **5. Question: A relevant body of work which has not been discussed in the paper is the one of predictive coding [1, 2]. In predictive coding, pixel-wise redundant spikes (and internal layer spikes) are removed, usually following some sort of minimization of the so-called free-energy [1]. I believe that the authors should at the very least cite this body of work and discuss differences between their work and some of the spiking counterparts of predictive coding [2], as both types of models aim at learning efficient visual representations through the removal of redundant spikes.**
>
> Response: Thank you very much for providing these excellent works. They indeed provide additional theoretical support for spike removal in our MI-TRQR. We will cite these references and revise the motivation (line 51) as follows in the final version:
>
> >Previous studies analyzing neural recordings from various monkey brain regions have shown that spike features are significantly less redundant with activity-dependent depression. **Predictive coding has also demonstrated the ability to learn efficient visual representations by removing pixel-wise redundant spikes [1,2].** Inspired by these works, we propose a parameter-free, plug-and-play MI-based Temporal Redundancy Quantification and Reduction (MI-TRQR) module, which is seamlessly integrated into the SNNs.
>
> **6. Question: I would kindly ask the authors to analyze the weaknesses/limitations of their work.**
>
> Response: Thank you for your suggestion. We will include a limitation section in the final version.
>
> >Although our method does not introduce extra cost during inference, it requires computing mutual information between spike features during training, resulting in increased computational cost and longer training time.
>
> **7. Question: I encourage the authors to modify the formatting of Sec. 4.1, as the text is not properly justified.**
>
> Response: Thank you for your suggestion. Due to space constraints, we arrange the text of Sec. 4.1 and Tab. 3 side by side. We will revise the formatting and improve layout quality in the final version.
>
> **References**
>
> [1] Bogacz, Rafal. "A tutorial on the free-energy framework for modelling perception and learning." Journal of mathematical psychology 76 (2017): 198-211.
>
> [2] N'dri, Antony W., et al. "Predictive Coding with Spiking Neural Networks: a Survey." arXiv preprint arXiv:2409.05386 (2024).

---

> > ### Author Response · Authors · 2025-08-04
> >
> > Thank you for your thoughtful comments and for the time you have invested in our work. We have posted our response and hope the above clarifications sufficiently addressed your concerns. If any questions remain unresolved, please let us know. We would be grateful to clarify further and are happy to discuss specific points.

---

> > ### Comment · Reviewer_nHVz · 2025-08-07
> >
> > Thank you for your response. It clarified sufficiently my concerns. I will therefore be maintaining my score.

---

### Official Review · Reviewer_CMhp · 2025-07-03

**Clarity:** 3
**Significance:** 3
**Originality:** 3
**Rating:** 4
**Confidence:** 3

**Summary:**

This paper proposes a novel Mutual Information-based Temporal Redundancy Quantification and Reduction approach (MI-TRQR) for energy-efficient SNNs. Extensive experimental results demonstrate that MI-TRQR achieves superior performance across neuromorphic data classification, static data classification, and time-series forecasting tasks.

**Questions:**

See in weakness.

**Ethical Concerns:**

["NO or VERY MINOR ethics concerns only"]

**Final Justification:**

The authors have addressed my questions. I will keep the positive review for this paper.

**Limitations:**

Yes

**Quality:**

3

**Strengths And Weaknesses:**

The topic is quite interesting and offers a novel perspective for improving SNN efficiency. The paper is well-written and well-structured.

However, I have a few suggestions and questions for the authors:

The ablation study shows that applying the MI-TRQR module after the last convolutional layer yields the best results. Could the authors provide further insight into this observation? This result is somewhat puzzling, as temporal redundancy likely exists in intermediate layers as well. In addition, have the authors considered applying MI-TRQR at multiple stages throughout the network?

Regarding the reported power consumption: how is the power measured? Is it obtained from real hardware or through simulation? If the latter, please clarify the simulation method and include the formulation used to estimate power consumption.

---

> ### Author Rebuttal · Authors · 2025-07-29
>
> Thank you very much for taking the time to review our paper.  Our responses to your questions are as follows:
>
> **1. Question: The ablation study shows that applying the MI-TRQR module after the last convolutional layer yields the best results. Could the authors provide further insight into this observation? This result is somewhat puzzling, as temporal redundancy likely exists in intermediate layers as well. In addition, have the authors considered applying MI-TRQR at multiple stages throughout the network?**
>
> Response: Thanks for your consideration. We address this concern in the following aspects:
>
> (1)	Why do we apply the MI-TRQR module after the last convolutional layer?
>
> Prior studies [1, 2] show the final convolutional layer features are most critical for correct classification, while earlier layers mainly learn reusable general representations and deeper layers become highly specific. Placing MI-TRQR after the last layer therefore helps the classifier directly filter out redundant spikes, and this decision can in turn dynamically guide the removal of redundancy in shallower layers. Its rigorous analysis and derivation are provided in Appendix B, a similar statement in line 160.
>
> (2)	Why we did not apply MI-TRQR at multiple stages?
>
> Redundancies at shallow and deep layers are not equivalent. Specifically, the redundancy identified in shallow layers may be useful in deep layers. Features that are considered useful in shallow layers may be useless in deep layers.
>
> (3) The computation of probability $p_{t,h,w}$ (Eq. 5) is spatiotemporally dependent, with a computational complexity of $\mathcal{O}(T)$ for global redundancy and $\mathcal{O}(T×H×W)$ for local redundancy. Therefore, inserting the module into earlier layers leads to higher computational cost. We provide the corresponding time of training an epoch based on Tab. 5. Due to time constraints, we only obtain the results of the module inserted after layers 6 and 8.
>
> |Method|Placement|Accuracy|Firing rate|Training time|
> |---|---|---|---|---|
> |PSN|-|82.3|14.47|24.1s|
> |MI-TRQR|After layer 4|82.8|11.52|597.3s|
> ||After layer 6|83.4|10.18|171.6s|
> ||After layer 6+8|83.5|9.74|222.6s|
> ||After layer 8|**84.0**|**8.35**|**51.9s**|
>
> We can see that adding the module to earlier layers introduces substantial computational cost, yet yields no gains in accuracy or firing rate reduction.
>
> **2. Question: Regarding the reported power consumption: how is the power measured? Is it obtained from real hardware or through simulation? If the latter, please clarify the simulation method and include the formulation used to estimate power consumption.**
>
> Response: The power consumption is obtained through simulation. The simulation method follows prior works [3,4], and the estimation formulation is provided in Eq. 11 (page 15). We show it as follows:
>
> $$
> E_{net}=E_{MAC} \cdot FL_{Conv}^{1} + E_{AC} \cdot T \cdot (\sum_{n=2}^{N}FL_{Conv}^{n}\cdot fr_{Conv}^{n}+ \sum_{m=1}^{M}FL_{FC}^{m}\cdot fr_{FC}^{m})
> $$
>
> where $FL_{Conv}^{1}$ is the FLOPs of first convolution layer, $FL_{Conv}^{n}$ and $FL_{FC}^{m}$ are the FLOPs of the $n$-th convolution layer and the $m$-th FC layer, $fr_{Conv}^{n}$, and $fr_{FC}^{m}$ are the firing rate of the $n$-th convolution layer and the $m$-th FC layer, respectively. $N$ and $M$ are the total number of convolution and FC layers. Following prior works [5,6], $E_{MAC}$=4.6pJ, $E_{AC}$=0.9pJ.
>
>
> **References**
>
> [1] Jie Hu, Li Shen, and Gang Sun, Squeeze-and-excitation networks. In Proceedings of the IEEE Conference on Computer Vision and Pattern Recognition, pages 7132–7141, 2018.
>
> [2] Bolei Zhou, Aditya Khosla, Agata Lapedriza, Aude Oliva, and Antonio Torralba. Learning deep features for discriminative localization. In 2016 IEEE Conference on Computer Vision and Pattern Recognition (CVPR), pages 2921–2929, 2016.
>
> [3] Hangchi Shen, Qian Zheng, Huamin Wang, and Gang Pan. Rethinking the membrane dynamics and optimization objectives of spiking neural networks. Advances in Neural Information Processing Systems, 37:92697–92720, 2024.
>
> [4] Xuerui Qiu, Rui-Jie Zhu, Yuhong Chou, Zhaorui Wang, Liang-jian Deng, and Guoqi Li. Gated attention coding for training high-performance and efficient spiking neural networks. In Proceedings of the AAAI Conference on Artificial Intelligence, volume 38, pages 601–610, 2024.
>
> [5] Mark Horowitz. 1.1 computing’s energy problem (and what we can do about it). In 2014 IEEE International Solid-State Circuits Conference Digest of Technical Papers (ISSCC), pages 10–14. IEEE, 2014.
>
> [6] Bojian Yin, Federico Corradi, and Sander M Bohté. Accurate and efficient time-domain classification with adaptive spiking recurrent neural networks. Nature Machine Intelligence, 3(10):905–913, 2021.

---

> > ### Author Response · Authors · 2025-08-04
> >
> > Thank you for your thoughtful comments and for the time you have invested in our work. We have posted our response and hope the above clarifications and the additional experiments sufficiently addressed your concerns. If any questions remain unresolved, please let us know. We would be grateful to clarify further and are happy to discuss specific points.

---

> > ### Comment · Reviewer_CMhp · 2025-08-08
> >
> > Good. The authors have addressed my question. I will keep the positive review for this paper.

---

> > > ### Author Response · Authors · 2025-08-08
> > >
> > > Thank you for your kind response. We are glad to hear that your question is addressed and truly appreciate your support and keeping the positive review.

---

### Official Review · Reviewer_5G1g · 2025-07-06

**Clarity:** 3
**Significance:** 3
**Originality:** 3
**Rating:** 3
**Confidence:** 3

**Summary:**

Based on earlier work that identifies redundancy among the learned spike features, the paper proposes a new nonlinear measure of distributional similarity - that of mutual information between the spike features across times. They measure the mutual information locally - between the same spike feature across timesteps, and globally across different features.  Based on these techniques, the authors propose a pruning (masking) technique for spikes and show that they can get higher accuracy and efficiency, along with lower spike rates on both event driven and static image data.

**Questions:**

I am happy to revise my score based on the answers to these questions, ordered by importance:
1. Reliability of the mutual information measuring technique
    1. Measuring mutual information is hard with sparse high dimensional data, since we just cannot have enough samples.  To the best of my understanding, how mutual information is measured is not mentioned anywhere in the equations, algorithms, main text or appendix. More details, including equations, and number of samples are required to complete this section.
    2. There need to be  some study done to verify that mutual information, as captured in this manuscript, is actually performing as expected.  For instance, the compelling reason for redundancy is for static images as they are duplicated across timesteps. However, we see similar impacts of MI-TRQR for both event driven and static datasets. Would we not have expected to see MI-TRQR be much more successful at reducing spikes for data with more redundancy (static duplicated images)
    3. What is the computational cost of calculating this mutual information? Since we have to do this during training, how much longer does training take now?
2. Ablation of the role of the spike normalization
    1. The authors do a good job of ablating the effect of local and global redundancy, and show that both help. However, I am curious about the role of spike normalization as shown in equation 5. This aims to suppress only the spikes occurring at the later layer, and if we applied the same ratios without masks to random spikes or to all spikes (without the mask, what would the accuracy be?
    2. If we had a static penalty of the number of spikes that increased with time steps as a regularizer, what would be the accuracy then at iso-firing rate?
3. Did we need to do MI based pruning? What would have happened if we swapped out MI with simpler techniques (perhaps even linear). Would spiking rate still have gone down? The ablations are done only against the whole network, without any spike pruning?
4. Did the resulting network actually have lesser redundant spike features? If you did the qualitative analysis and/or measured MI for the original and new network, would they be lesser correlated, perhaps even linearly?

Good to have, but lower impact on rating.
    1. Why Bernoulli? Why not threshold against a value?
    2. I don’t understand the placement experiment (line 208-211) Can the authors explain why if the reduction in spikes is placed after only at the last layer (layer 8 IIUC) the firing rate is suppressed more? Why not across all layers? I would have intuitively expected more redundancy in lower level features than high level features.
    3. [Minor] Algorithm 1 is not referenced at any point in the main text.
    4. [Minor] Tables should occur on the same page as they are referenced, ideally

**Ethical Concerns:**

["NO or VERY MINOR ethics concerns only"]

**Limitations:**

yes

**Quality:**

3

**Strengths And Weaknesses:**

Strengths
1. The experiment settings are comprehensive - they cover direct training and conversion, and both event driven and static image datasets.
2. The paper is really clearly written, with a sensible flow.
3. The metric of mutual information makes intuitive sense, as information centric techniques can reveal correlations that simple techniques such as Pearson correlation cannot
4. The results show good improvement in efficiency, and it is measured both by firing rate and power

Weaknesses
See Questions

---

> ### Author Rebuttal · Authors · 2025-07-28
>
> Thank you very much for taking the time to review our paper. Our responses to your questions are as follows:
>
> **1. Reliability of the mutual information measuring technique**
>
> **1.1. Question: Details of mutual information.**
>
> Response:
>
> (1). We provide the equation of mutual information (MI) in Appendix A (line 452).
>
> (2). The MI calculation is implemented using the **TorchMetrics** package (line 525). The implementation code is:
>
> ```python
> MI_value = torchmetrics.functional.clustering.mutual_info_score(S1, S2)
> ```
>
> where S1 and S2 are two spiking features. $S1,S2\in\lbrace0,1\rbrace^{C×H×W}$ and $S1,S2\in\lbrace 0,1\rbrace ^C$ when computing global and local redundancy. MI_value is a scalar, indicating the MI between $S1$ and $S2$.
>
> (3) It does need many samples for the computation of MI. The MI of two random variables is a measure of the mutual dependence between the two variables. Therefore, the calculation of MI only needs two spike features.
>
> **1.2. Question: Impacts of MI-TRQR for both event-driven and static datasets.**
>
> Response: The reviewer points out the similar impacts for both event-driven and static datasets. We believe the reviewer is referring to the reduction in energy consumption on CIFAR10-DVS (**37.50%** with T=4) and ImageNet (**24.82%** on ResNet50). We argue that the CIFAR10-DVS is a small dataset (10,000 samples in 10 classes), while ImageNet is a large-scale dataset (about 1.3 million images in 1,000 classes). Therefore, they are not directly comparable. We also show the impact on CIFAR10 (60,000 samples in 10 classes) in Appendix C.2. MI-TRQR reduces energy consumption by **73.48%**. Therefore, our MI-TRQR achieves the expected results, in that it is much more successful at reducing spikes for data with more redundancy (static duplicated images).
>
> **1.3. Question: The computational cost of calculating mutual information.**
>
> Response: We show the time of training an epoch using different methods with a single NVIDIA RTX 3090 GPU:
>
> |Dataset|Method|Network|Local|Global|Time|
> |---|---|---|---|---|---|
> |CIFAR10-DVS|PSN|VGG|-|-|24.1s|
> ||MI-TRQR|VGG|-|$\checkmark$|24.8s|
> |||VGG|$\checkmark$|-|51.1s|
> |||VGG|$\checkmark$|$\checkmark$|51.9s|
> |ImageNet|PSN|VGG|-|-|68.9min|
> ||MI-TRQR|VGG|-|$\checkmark$|70.4min|
> |||VGG|$\checkmark$|-|78.5min|
> |||VGG|$\checkmark$|$\checkmark$|79.0min|
>
> **2. Ablation of the role of the spike normalization**
>
> **2.1. Question: The same ratios without masks to random spikes or to all spikes**
>
> Response: Thanks for your profound thinking.
>
> (1) The mask is one concrete strategy to remove redundant spikes; other mechanisms could also be used. The reason why we use the mask is that matrix calculation is more efficient in current machines.
>
> (2) When only global redundancy is used to remove redundant spikes, the removal within $S_t\in\lbrace0,1\rbrace^{C×H×W}$ at timestep $t$ is essentially random.
>
> (3) We apply the same ratios to all spikes. Specifically, after obtaining the mask with $p_{t,h,w}$ (Eq. 5), we shuffle all elements to generate a random mask for spike removal.
>
> |Method|Shuffle|Accuracy|Firing rate|
> |---|---|---|---|
> |PSN|-|82.3|14.47|
> |MI-TRQR|$\checkmark$|83.4|8.51|
> ||-|84.0|8.35|
>
> We observe that both the MI-based ratio and the removed spikes are crucial.
>
> **2.2 Question: Static penalty of the number of spikes.**
>
> Response: We use different static penalty factors:
>
> |Method|$t$=1|$t$=2|$t$=3|$t$=4|Accuracy|Firing rate|
> |---|---|---|---|---|---|---|
> |PSN|-|-|-|-|82.3|14.47|
> |Static penalty|0|0.1|0.2|0.3|83.0|10.83|
> ||0|0.15|0.3|0.45|82.9|9.55|
> ||0|0.2|0.4|0.6|82.8|10.61|
> |MI-TRQR|-|-|-|-|**84.0**|**8.35**|
>
> Static penalties are indeed effective at reducing redundant spikes. Notably, MI-TRQR achieves better results on accuracy and firing rate.
>
> **3. Question: Do MI based pruning? Swap out MI with simpler techniques (perhaps even linear). Would spiking rate still have gone down? The ablations are done only against the whole network, without any spike pruning?**
>
> Response:
>
> (1) Spike removal and pruning serve distinct purposes: pruning discards low-impact weights, whereas our method removes redundant spikes from features. This distinction allows the two to be applied together in a complementary manner.
>
> (2) We swap out MI with simpler techniques, such as Euclidean similarity (via 1/(1+Euclidean distance)), Pearson correlation, and Cosine similarity.
>
> |Method|Technique|Accuracy|Firing rate|
> |---|---|---|---|
> |PSN|-|82.3|14.47|
> |Absolute distance|Euclidean|82.4|11.74|
> |Linear similarity|Pearson|82.5|10.92|
> |Feature direction|Cosine|82.9|9.59|
> |MI-TRQR|MI|**84.0**|**8.35**|
>
> We can see that the spiking rate still goes down using simpler techniques. Notably, MI-TRQR performs better.
>
> (3) Ablation with spike pruning.
>
> We multiply $p_{t,h,w}$ (Eq. 5) by an additional factor $a$, and the results of different factors are:
>
> |Method|a|Accuracy|Firing rate|
> |---|---|---|---|
> |PSN|-|82.3|14.47|
> |MI-TRQR|0.5|84.5|10.66|
> ||1|84.0|8.35|
> ||2|82.7|8.16|
>
> We can see that factor $a$ affects the balance between accuracy and firing rate.
>
> **4. Question: Lesser redundant spike features? Lesser correlated, perhaps even linearly?**
>
> Response:
>
> (1) Yes. The resulting network actually has fewer redundant spike features. We show the MI for the original and new networks on CIFAR10-DVS and ImageNet in Fig. 1 and Fig. 4, respectively. Specifically, the spike features of the original network at four timesteps on CIFAR10-DVS are placed in Fig. 1(a), and their MI matrix is placed in Fig. 1(b). The spike features of the new network at four timesteps on CIFAR10-DVS are placed in Fig. 1(c), and the measured MI is placed in Fig. 1(d). We show the results on ImageNet using the same layout in Fig. 4. As shown in Fig. 1, MI-TRQR reduces the MI on CIFAR10-DVS by about **24%** (from 0.1507 to 0.1134) and on ImageNet by about **79%** (from 0.5837 to 0.1081). The related statements about Fig. 1 are in lines 28, 50, and 62.
>
> (2) Using the results in Question 3(3), we compute the MI of spike features (Fig. 1) using different $a$.
>
> $a$=0.5:
>
> |||||
> |---|---|---|---|
> |0.6167|0.1378|0.1201|0.1122|
> |0.1378|0.6056|0.2203|0.1901|
> |0.1201|0.2203|0.6033|0.2081|
> |0.1122|0.1901|0.2081|0.6003|
>
> $a$=2:
>
> |||||
> |---|---|---|---|
> |0.5097|0.1129|0.0993|0.0922|
> |0.1129|0.5064|0.1875|0.1801|
> |0.0993|0.1857|0.5009|0.2010|
> |0.0922|0.1801|0.2010|0.4987|
>
> We can see that it is not linear.
>
> **5. Question: Why Bernoulli? Why not threshold against a value?**
>
> Response:
>
> (1) Bernoulli-based masks admit a clearer theoretical interpretation. In contrast, a threshold lacks such grounding.
>
> (2) Bernoulli-based masks remove redundant spikes in a data-dependent manner and avoid introducing extra hyperparameters. The specific threshold is a hyperparameter that requires extensive studies to find the optimal configuration on different datasets.
>
> (3) To further verify the effectiveness of our method, we define a global redundancy-based threshold for pixel-wise spike removal. Specifically, for the feature $S_t\in\lbrace0,1\rbrace^{C\times H\times W} (t>1)$, we compute the global redundancy $p_t^g$ (Eq. 3) and the local redundancy $p_{t,h,w}^l$ (Eq. 4). If $p_{t,h,w}^l>p_t^g$, the spike feature $S_{t,h,w}\in\lbrace0,1\rbrace^C$ is set to 0.
>
> |Method|Accuracy|Firing rate|
> |---|---|---|
> |PSN|82.3|14.47|
> |MI-TRQR with threshold|82.6|8.97|
> |MI-TRQR with Bernoulli| **84.0**|**8.35**|
>
> These results demonstrate that our method with Bernoulli performs better.
>
> **6. Question: The placement experiment (line 208-211). Why not across all layers?**
>
> Response: Thanks for your consideration.
>
> (1)	Why is our MI-TRQR placed after only the last layer?
>
> Prior studies [1, 2] show the final convolutional layer features are most critical for correct classification, while earlier layers mainly learn reusable general representations and deeper layers become highly specific. Placing MI-TRQR after the last layer therefore helps the classifier directly filter out redundant spikes, and this decision can in turn dynamically guide the removal of redundancy in shallower layers. Its rigorous analysis and derivation are provided in Appendix B, a similar statement in line 160.
>
> (2)	Why not across all layers?
>
> First, redundancies at shallow and deep layers are not equivalent. Specifically, the redundancy identified in shallow layers may be useful in deep layers. Features that are considered useful in shallow layers may be useless in deep layers.
>
> (3) The computation of probability $p_{t,h,w}$ (Eq. 5) is spatiotemporally dependent, with a computational complexity of $\mathcal{O}(T)$ for global redundancy and $\mathcal{O}(T×H×W)$ for local redundancy. Therefore, inserting the module into earlier layers leads to higher computational cost. We provide the corresponding time of training an epoch based on Tab. 5. Due to time constraints, we only obtain the results of the module inserted after layers 6 and 8.
>
> |Method|Placement|Accuracy|Firing rate|Training time|
> |---|---|---|---|---|
> |PSN|-|82.3|14.47|24.1s|
> |MI-TRQR|After layer 4|82.8|11.52|597.3s|
> ||After layer 6|83.4|10.18|171.6s|
> ||After layer 6+8|83.5|9.74|222.6s|
> ||After layer 8|**84.0**|**8.35**|**51.9s**|
>
> We can see that adding the module to earlier layers introduces substantial computational cost, yet yields no gains in accuracy or firing rate reduction.
>
> **7. Question: Algorithm 1 is not referenced.**
>
> Response: Algorithm 1 is referenced at line 159.
>
> **8. Question: Tables should occur on the same page as they are referenced, ideally.**
>
> Response: Thanks for your advice. We will try our best to improve the layout.
>
> **References**
>
> [1] Squeeze-and-excitation networks. CVPR 2018
>
> [2] Learning deep features for discriminative localization. CVPR 2016

---

> > ### Author Response · Authors · 2025-08-04
> >
> > Thank you for your thoughtful comments and for the time you have invested in our work. We have posted our response and hope the above clarifications and the additional experiments sufficiently addressed your concerns. If any questions remain unresolved, please let us know. We would be grateful to clarify further and are happy to discuss specific points.

---

> > > ### Author Response · Authors · 2025-08-06
> > >
> > > We would like to make two clarifications regarding our previous rebuttal.
> > >
> > > ---
> > >
> > > 1. There is a minor typo in **Response (3)** to **Question 1.1** of our rebuttal. The correct statement should be:
> > >
> > > > It does **NOT** need many samples for the computation of MI. The MI of two random variables is a measure of the mutual dependence between the two variables. Therefore, the calculation of MI only needs two spike features.
> > >
> > > We apologize for any confusion this may have caused.
> > >
> > > ---
> > >
> > > 2. Our description in **Response (2)** to **Question 4** may be unclear. We apologize and provide the following clarification.
> > >
> > > By adjusting the value of $a$ (see Question 3(3)), we control the sparsity of spike features and analyze how temporal correlation changes. Our observations indicate that while the correlation decreases as more spikes are removed, the reduction is **not linear**.

---

> > ### Comment · Reviewer_5G1g · 2025-08-07
> >
> > Question regarding the use of torchmetrics mutual information calculation:
> >
> > https://lightning.ai/docs/torchmetrics/stable/clustering/mutual_info_score.html states that this metric is supposed to be used for two vectors that represent cluster assignments. I do not think that binary spiking vectors directly can be considered as cluster assignments for this purpose. Can the authors confirm that their mutual information calculation is correct please?

---

> ### Author Response · Authors · 2025-08-07
>
> Thanks very much for your question. We confirm that the mutual information (MI) calculation is correct, and we clarify this from three perspectives:
>
> (1) MI is defined and analyzed by Claude Shannon [1] and termed later by Robert Fano [2]. MI measures the statistical dependency between **two discrete variables**. Its computation only requires the inputs to be discrete variables and does not assume that they must represent cluster assignments. MI has been widely applied in various domains [3, 4], including in SNNs to quantify the information content in spike features [5, 6].
>
> (2) In our case, MI is particularly suitable for quantifying redundancy between binary spiking vectors (0/1 sequences), which are **a form of discrete variables**. As discussed in Response (2) to Question 3, MI is more powerful than alternative similarity metrics.
>
> (3) We compute MI using both the TorchMetrics and NPEET packages, and obtain **consistent results**. This further confirms the correctness of our implementation.
>
> ---
>
> **References**
>
> [1] Claude Elwood Shannon. A mathematical theory of communication. The Bell System Technical Journal, 27(3):379–423, 1948.
>
> [2] Kreer, J. G. (1957). "A question of terminology". IRE Transactions on Information Theory. 3 (3): 208. doi:10.1109/TIT.1957.1057418
>
> [3] Vergara J R, Estévez P A. A review of feature selection methods based on mutual information. Neural computing and applications, 24(1): 175-186, 2014.
>
> [4] Batina L, Gierlichs B, Prouff E, et al. Mutual information analysis: a comprehensive study. Journal of Cryptology, 24(2): 269-291,2011.
>
> [5] Yuhan Zhang, Xiaode Liu, Yuanpei Chen, Weihang Peng, Yufei Guo, Xuhui Huang, and Zhe Ma. Enhancing representation of spiking neural networks via similarity-sensitive contrastive learning. In Proceedings of the AAAI Conference on Artificial Intelligence, volume 38, pages 16926–16934, 2024.
>
> [6] Shikuang Deng, Yuhang Wu, Kangrui Du, and Shi Gu. Spiking token mixer: An event-driven friendly former structure for spiking neural networks. In Advances in Neural Information Processing Systems, volume 37, pages 128825–128846, 2024.

---

### Decision · Program_Chairs · 2025-09-17

**Decision:**

Accept (poster)

**Comment:**

##  Summary
This paper proposes MI-TRQR, a novel, parameter-free, plug-and-play module designed to quantify and reduce temporal redundancy in Spiking Neural Networks (SNNs) to improve energy efficiency and accuracy. The method uses mutual information (MI) to measure redundancy between discrete spike features across timesteps on both local (pixel-level) and global (spatial feature-level) scales. Based on the quantified redundancy, MI-TRQR applies probabilistic masking to remove redundant spikes and recalibrates the final representation. Extensive experiments demonstrate that MI-TRQR achieves sparser spike firing, reduced energy consumption, and improved performance across diverse SNN architectures and tasks including neuromorphic data classification, static image classification, and time-series forecasting.

## Strengths
- **Novelty:** Introduces a mutual information based approach to quantify temporal redundancy in spike features, a fresh perspective in SNN efficiency optimization.
- **Effectiveness:** Demonstrates simultaneous gains in accuracy and substantial reductions in spike rates and energy consumption across multiple datasets and architectures.
- **Comprehensive evaluation:** Covers event-driven and static image datasets, including large-scale ImageNet, with ablation studies on local/global redundancy and spike normalization.
- **Clarity:** Well-written and logically structured manuscript with clear explanations and detailed ablations.
- **Compatibility:** The MI-TRQR module is parameter-free and only used during training, preserving inference efficiency and compatibility with existing SNN frameworks.



## Weaknesses and Missing Elements
- **Mutual information estimation concerns:** Questions raised regarding the validity of the MI calculation method on sparse binary spike data, requiring careful justification and validation.
- **Computational cost:** MI calculation introduces additional training overhead; while inference cost is unaffected, training time increase may be significant.
- **Theoretical analysis:** Limited theoretical discussion on why MI is especially suitable for binary spike features and how weight recalibration compensates for spike removal.
- **Module placement:** Ablation shows best results when MI-TRQR is applied after the last convolutional layer, but deeper analysis on redundancy across layers and multi-stage application is limited.
- **Comparison to related methods:** Lack of direct experimental comparison to other redundancy reduction or pruning methods in SNN literature reduces clarity on relative advantage.
- **Limited discussion of limitations:** Initial submission lacks a dedicated section on method limitations and potential bottlenecks.


## Discussion and Rebuttal Summary

### Reviewers and Their Key Concerns:
- **Reviewer 5G1g:**
  - Raised a critical question on the correctness of the mutual information estimation approach for sparse high-dimensional binary spike vectors.
  - Requested more detailed explanation of MI calculation, its reliability, and sample requirements.
  - Asked about differences in effectiveness between static and event-driven datasets.
  - Inquired about computational cost and efficiency impact during training.
  - Questioned the role of spike normalization and whether simpler similarity measures would suffice.
  - Requested ablations on MI-based pruning and evidence that redundancy is actually reduced.

- **Reviewer CMhp:**
  - Asked for clarification on why MI-TRQR module is applied only after the last convolutional layer and if multi-stage application was considered.
  - Requested details on power consumption measurement methodology (simulation vs. hardware).
  - Appreciated clarity and structure but desired more insight on module placement and efficiency metrics.

- **Reviewer nHVz:**
  - Requested clarification on weight recalibration mechanics and its theoretical basis.
  - Raised concern that MI-TRQR operates on continuous quantities, potentially limiting fully spike-based deployment.
  - Suggested citing and discussing related predictive coding literature.
  - Requested analysis of weaknesses/limitations.
  - Asked for formatting improvements.

- **Reviewer wE4j:**
  - Expressed concern about the theoretical analysis of MI effectiveness for binary spikes.
  - Pointed out potential increase in training complexity and energy cost due to MI calculations.
  - Questioned the rationality of using first timestep as reference in MI calculation.
  - Requested ablation studies on reweighting effectiveness and comparison to other redundancy reduction methods.
  - Noted limited performance gains on time-series forecasting and asked for statistical robustness of results.

### Authors’ Responses:
- Provided detailed explanation and equations for MI calculation using TorchMetrics and NPEET libraries, confirming correctness and consistency.
- Argued MI is suitable for discrete binary variables and demonstrated effectiveness empirically.
- Clarified computational overhead is limited to training phase; inference remains efficient.
- Explained rationale for using first timestep as reference based on temporal information concentration (TIC) phenomenon and ensemble learning perspectives.
- Conducted ablation studies showing spike normalization and reweighting improve accuracy and spike rate reduction.
- Compared MI-TRQR to simpler similarity metrics and redundancy reduction methods, showing MI-based method performs best.
- Presented module placement ablations showing best trade-off when applied after last convolutional layer due to computational cost and feature specificity.

### Final Decision:
The authors addressed major concerns with detailed explanations, additional experiments, and clarifications that validate the MI calculation and demonstrate practical benefits. Some questions about computational cost and theoretical framing remain but are reasonably mitigated by the fact that overhead is training-only. The paper presents a meaningful contribution toward energy-efficient SNNs by tackling temporal redundancy with a novel MI-based approach. The empirical results are convincing and robust across datasets and architectures. Given the borderline opinions of reviewers and the authors’ thorough rebuttal, this submission merits acceptance conditioned on addressing minor outstanding points in the final version.